# Beyond Matryoshka: Revisiting Sparse Coding for Adaptive Representation

Tiansheng Wen [* 1 2]   Yifei Wang [* 3]   Zequn Zeng [1]   Zhong Peng [1]   Yudi Su [1]   Xinyang Liu [1]   Bo Chen [1]
Hongwei Liu [1]   Stefanie Jegelka [3 4]   Chenyu You [2]

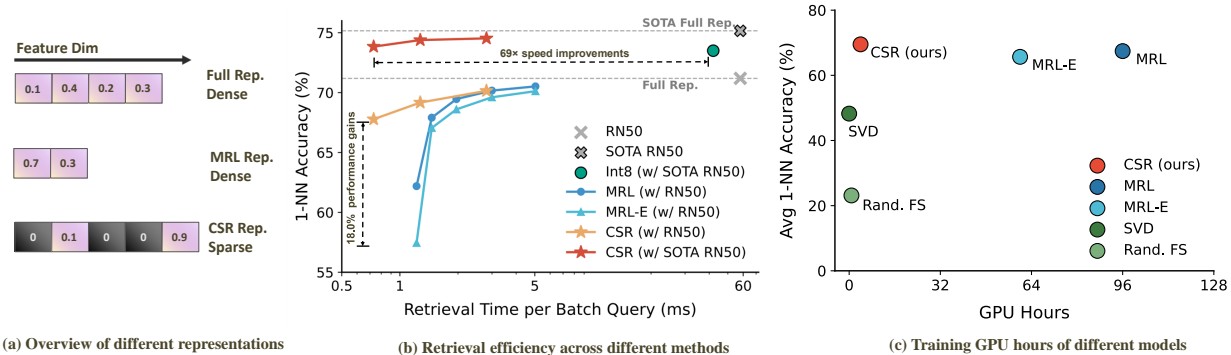

(a) Overview of different representations   (b) Retrieval efficiency across different methods   (c) Training GPU hours of different models

*Figure 1.* Overview of our proposed method. (a) Illustrative comparison between standard embeddings (dense, long) and two different compression schemes: Matryoshka representations (MRL) (Kusupati et al., 2022) with short length and our Contrastive Sparse Representation (CSR) based on sparsification. (b) Comparison of retrieval accuracy and time of different methods on ImageNet with GPUs. For CSR, we present results with the SOTA RN50 backbone from Wightman (2019) as well as the same RN50 backbone from Kusupati et al. (2022) for a fair comparison. Compared to MRL and int8 quantification (Quant Int8) methods, our sparse embedding approach CSR attains the best retrieval accuracy (very close to full representations) while being much more efficient in retrieval time, using sparse matrix multiplication on GPU. (c) Training GPU hours of CSR compared to baseline methods, where we outperform MRL on average 1-NN accuracy with much less training time.

## Abstract

Many large-scale systems rely on high-quality deep representations (embeddings) to facilitate tasks like retrieval, search, and generative modeling. Matryoshka Representation Learning (MRL) recently emerged as a solution for adaptive embedding lengths, but it requires full model retraining and suffers from noticeable performance degradations at short lengths. In this paper, we show that *sparse coding* offers a compelling alternative for achieving adaptive representation with minimal overhead and higher fidelity. We propose **Contrastive Sparse Representation** (**CSR**), a method that sparsifies pre-trained embeddings into a high-dimensional but *selectively activated* feature space. By leveraging lightweight autoencoding and task-aware contrastive objectives, CSR preserves semantic quality while allowing flexible, cost-effective inference at different sparsity levels. Extensive experiments on image, text, and multimodal benchmarks demonstrate that CSR consistently outperforms MRL in terms of both accuracy and retrieval speed—often by large margins—while also cutting training time to a fraction of that required by MRL. Our results establish sparse coding as a powerful paradigm for adaptive representation learning in real-world applications where efficiency and fidelity are both paramount. Code is available at this https URL.

[*]Equal contribution [1]National Key Laboratory of Radar Signal Processing, Xidian University, Xi'an, China [2]Stony Brook University, New York, USA [3]MIT CSAIL, MA, USA [4]TU Munich. Correspondence to: Bo Chen <bchen@mail.xidian.edu.cn>, Hongwei Liu <hwliu@xidian.edu.cn>.

*Proceedings of the 42nd International Conference on Machine Learning*, Vancouver, Canada. PMLR 267, 2025. Copyright 2025 by the author(s).

## 1. Introduction

Representation learning is at the core of deep learning (LeCun et al., 2015) and high-quality representations of inputs (*e.g.*, image, text) empower numerous large-scale systems, including but not limited to search engines, vector databases,

and retrieval-augmented generative AI (Lewis et al., 2020). However, the rapid growth in data volume poses significant challenges for latency-sensitive applications. It is thus desirable to develop representations of adaptive inference cost that can best trade-off between accuracy and inference speed.

Recently, a class of methods called Matryoshka Representation Learning (MRL) (Kusupati et al., 2022) has drawn a lot of attention and is now officially supported in the latest OpenAI and Google's Gemini text embedding APIs (OpenAI, 2024; Lee et al., 2024b) with millions of users and applications. The idea if MRL is to train an ensemble of representations truncated at different lengths (*e.g.*, from 8 to 2048) through joint multi-task training. However, MRL deviates from standard representation learning and requires full parameter updates to the backbone; the joint training also inevitably sacrifices the quality of representations at a noticeable margin (*e.g.*, 5% drop of top-1 accuracy on ImageNet at full representation length). These limitations render MRL a costly and lossy method for adaptive representation.

In this paper, we revisit sparse coding (Lee et al., 2006) as **a much faster, lightweight, and high-fidelity** approach to achieve adaptive representation. As illustrated in Figure 1(a), instead of truncating the representation length as in MRL, we leverage sparse vectors and sparse matrix factorization to attain computational efficiency. Specifically, we sparsify a full representation at different levels (characterized by $K$, the number of activated neurons). We find that a few numbers of activated neurons (*e.g.*, 4 to 16) can preserve the performance of a much longer dense representation (*e.g.*, 2048 dimensions). This is in sharp contrast to MRL embeddings whose quality deteriorates a lot at such extremely short lengths (>10% drop). Therefore, sparse features using sparse vector formats can be stored efficiently with only a few activated neurons. With the help of sparse matrix factorization (with native GPU support in modern deep learning libraries such as PyTorch)[1], these sparse embeddings can be used for retrieval tasks at a much higher speed with a complexity order of $\mathcal{O}(K)$, where $K$ is very small. In comparison, MRL requires a longer length of representation (*e.g.* 256) to attain similar accuracy (if possible), leading to extra slower inference speed. As shown in Figure 1(b), MRL is inferior to our method in terms of both accuracy and retrieval time by a significant margin.

Another key advantage of sparse features is that they eliminate the need to retrain the entire network. In contrast, MRL—Kusupati et al. (2022) noted—performs poorly unless full-parameter tuning. However, many existing foundation models, such as the multimodal representations in CLIP (Radford et al., 2021) and the text embeddings in NV-Embed

---

[1]PyTorch's native sparse vector library can be found at `https://pytorch.org/docs/stable/sparse.html`.

(Lee et al., 2024a), are pre-trained as single representations on massive Internet-scale data. Fine-tuning these models would be prohibitively expensive and would prevent leveraging pre-trained open weights. Leveraging recent advances in training sparse autoencoders (SAEs) (Cunningham et al., 2023; Gao et al., 2024), we can train a lightweight 2-layer MLP module for sparsifying pre-trained embeddings within a very short period of time (*e.g.*, half of an hour on ImageNet with a single GPU), which is of orders of magnitude faster than MRL, as shown in Figure 1(c).

These pieces of evidence on accuracy, retrieval time, and training time show that sparse features are strong alternatives to MRL methods for producing high-fidelity and computationally efficient representations with a lightweight module and training cost. Our proposed method, **Contrastive Sparse Representation Learning (CSR)**, combines contrastive retrieval and reconstructive autoencoding objectives to preserve the original feature semantics while better tailing it down to the retrieval tasks. We evaluate CSR on a range of standard embedding benchmarks, from image embedding, text embedding, to multimodal embeddings, and compare it against various state-of-the-art efficient embedding models. Extensive experiments show that CSR consistently outperforms MRL and its variants by significant margins in terms of both accuracy and efficiency. Notably, under the same compute budget, CSR rivals MRL's performance by 9%, 15%, and 7% on ImageNet classification, MTEB text retrieval, and MS COCO retrieval, respectively. Our main contributions are:

- We propose sparse coding as an alternative approach to adaptive representation learning and demonstrate its numerous advantages over the MRL approach in terms of fidelity, retrieval cost, and training cost.

- We introduce an effective learning method for sparse adaptive representation, **Contrastive Sparse Representation (CSR) Learning**. It combines a task-specific sparse contrastive learning loss with a reconstructive loss to maintain overall embedding quality. This generic design consistently improves performance across different tasks like classification and retrieval.

- We conduct a detailed analysis of CSR, examining various factors and providing a fair comparison with MRL in terms of retrieval time and accuracy. We further validate CSR's effectiveness across real-world domains and benchmarks, where it achieves competitive performance against heavily trained state-of-the-art MRL models with significantly lower computational costs. On the inference side, CSR delivers a **69× speedup** on ImageNet1k 1-NN tasks without compromising performance compared to quantization-based approaches.

## 2. Related Work

**Adaptive Representation Learning.** Recent research has increasingly focused on learning *adaptive representations* that cater to multiple downstream tasks with diverse computational requirements. Early efforts explored context-based architectural adaptations (Kim & Cho, 2020), dynamic widths and depths in BERT (Hou et al., 2020), and random layer dropping during training to improve pruning robustness (Fan et al., 2019). More recently, Matryoshka Representation Learning (Kusupati et al., 2022) introduced a novel technique for creating flexible, nested substructures within embeddings, enabling fine-grained control over the trade-off between latency and accuracy. This concept has since been extended to various modalities and applications, including large language models (OpenAI, 2024; Nussbaum et al., 2024; Yu et al., 2024), diffusion models (Gu et al., 2023), and multimodal models (Cai et al., 2024; Hu et al., 2024). Other works have further explored token reduction in image and video processing (Yan et al., 2024b; Duggal et al., 2024).

Despite these advances, existing methods often do not fully harness the capabilities of large foundation models, highlighting the need for more effective compression strategies. Our proposed *sparse compression* methodology addresses this gap by providing a lightweight, plug-and-play solution that can be readily applied on top of any foundation model – significantly reducing computational overhead while preserving representational quality.

**Sparse Coding.** Sparse coding serves as a powerful technique for compressing high-dimensional signals and extracting salient features (Wright et al., 2010; Zhang et al., 2015), with learned sparse representations often providing additional computational benefits and robustness (You et al., 2024; 2025). Prior work has induced sparsity through modifications to model design or training protocols, including modifications to attention mechanisms (Correia et al., 2019), applying Bayesian standard Gamma priors (Duan et al., 2024a;b; Hu et al., 2025), incorporating discrete sparse concept layers (Koh et al., 2020; Xie et al., 2025), and promoting sparse activations in large language models (Mirzadeh et al., 2023; Zhang et al., 2024). However, training state-of-the-art foundation models from scratch under these sparsity constraints has proven challenging (Elhage et al., 2022), limiting their current applicability.

Meanwhile, Sparse Autoencoders have achieved notable success in improving the interpretability of foundation models (Cunningham et al., 2023; Yan et al., 2024a), primarily because they uncover semantic information by mapping high-dimensional data onto lower-dimensional subspaces (Cunningham et al., 2023). Building on these insights – and harnessing the inherent advantages of sparse coding – we investigate how SAEs can be further developed to learn adaptive representations with high efficiency, expanding their applicability to a wider range of tasks.

## 3. Method

Our proposed framework, Contrastive Sparse Representation learning (**CSR**), is illustrated in Figure 2. Starting from a pre-trained embedding $v \in \mathbb{R}^d$, we project it into a sparse representation space $\mathbb{R}^h$, selectively activating the most relevant dimensions for adaptive representation learning. We then regularize this hidden space using a reconstruction-based sparse compression loss (Section 3.2.1). Additionally, with theoretical motivations and guarantees provided by (Wang et al., 2024), we introduce a non-negative contrastive loss to expand model capacity and feature identifiability. (Section 3.2.2)

### 3.1. Preliminaries

**Problem Formulation.** For simplicity, we first introduce our framework in the context of a classification task. Let $\mathcal{D}_{db}^N = \{(x_i, y_i)_{i=1}^N\}$ be a training dataset of size $N$, where $x_i \in \mathcal{X}$ are an input sample and $y_i \in \mathcal{Y}^L$ are corresponding labels with $L$ classes, We obtain an embedding $v = f(x; \theta_f) : \mathcal{X} \to \mathbb{R}^d$. We can apply exact $\ell_2$-based $k$-nearest neighbor (KNN) search for classification, which has $\mathcal{O}(dN)$ complexity. In practice, KNN often employs high-dimensional embeddings (*i.e.* $d = 4096$) to achieve stronger performance, but at the cost of increased computational latency. Our goal is to learn a more compact representation $v' \in \mathbb{R}^m$ (where $m \ll d$) that balances accuracy and query latency. This shortened embedding can also benefit other downstream tasks such as retrieval and clustering.

**Matryoshka Representation Learning (MRL).** MRL (Kusupati et al., 2022) simultaneously optimizes embeddings at multiple dimensions, as illustrated in Figure 2, to produce representations of variable size. Specifically, let $\mathcal{M}$ be a set of target embedding sizes. For each $m \in \mathcal{M}$, MRL applies an additional linear classifier to the first $m$ dimensions of the embedding vector, $v_{1:m} \in \mathbb{R}^m$. This design ensures each truncated representation is explicitly trained via the final loss. Formally, the MRL objective is

$$\mathcal{L}_{\text{MRL}} = \sum_{m \in \mathcal{M}} c_m \mathcal{L}_{\text{CE}} \left( \boldsymbol{W}^{(m)} \cdot f(x_i; \theta_f)_{1:m}; y_i \right), \quad (1)$$

where $\boldsymbol{W}^{(m)} \in \mathbb{R}^{L \times m}$ is the linear classifier weights corresponding to $v_{1:m}$. Each loss term is scaled by a non-negative coefficient $\{c_m \geq 0\}_{m \in \mathcal{M}}$. The multi-granularity arises from selecting dimensions in $\mathcal{M}$, whose size is constrained to at most $\log(d)$, that is, $|\mathcal{M}| \leq \lfloor \log(d) \rfloor$. For example, Kusupati et al. (2022) choose $\mathcal{M} = \{8, 16, \ldots, 1024\}$ as the nesting dimensions.

## 3.2. Contrastive Sparse Representation

As discussed in Section 1, MRL (Equation 1) faces two key constraints: it requires (full) training of the backbone parameters $\theta_f$ and its performance often deteriorates a lot under small hidden dimensions. To overcome these limitations, we propose a new methodology that relies on the computational efficiency of *sparse vectors* for efficient retrieval. The method, named Contrastive Sparse Representation (CSR), learns a simple one-layer sparse module on top of *frozen* pretrained embedding models (with full representation size, *e.g.*, 2048) that maps dense embeddings to highly sparse embeddings with a small number of active (i.e., non-zero) dimensions (*e.g.*, 32). As a result, CSR not only saves a lot training effort, but also allow using sparse matrix multiplication at inference time to accelerate retrieval significantly. Below, we outline how we train the CSR module through a combination of sparse autoencoding (Section 3.2.1) and sparse contrastive learning (Section 3.2.2).

### 3.2.1. SPARSE AUTOENCODING

Autoencoding is a long-standing unsupervised objective that extract salient features that could preserve the original data the most a reconstruction objective. In CSR, we aim at compressing dense embeddings to sparse vectors for efficient sparse retrieval while retaining most of the useful information. To achieve this goal, we adopt sparse autoencoders due to their ability to scale with large data and restore feature semantics (Cunningham et al., 2023; Yan et al., 2024a).

**Sparse Autoencoders (SAEs).** SAEs (Makhzani & Frey, 2013; Cunningham et al., 2023; Gao et al., 2024; Yan et al., 2024a) aim to extract a sparse representation $z_k$ by learning to reconstruct the dense feature from $z_k$. Specifically, given a pretrained dense embedding $v := f(x) \in \mathbb{R}^d$ as the input, we apply a TopK SAE (Gao et al., 2024) with the following autoencoding process:

$$z_k := \sigma^+(\mathrm{TopK}(\boldsymbol{W}_{\mathrm{enc}}(f(x) - \boldsymbol{b}_{\mathrm{pre}}) + \boldsymbol{b}_{\mathrm{enc}})), \quad (2)$$

$$\widehat{f(x)}_k := \boldsymbol{W}_{\mathrm{dec}}z_k + \boldsymbol{b}_{\mathrm{pre}}, \quad (3)$$

where $\boldsymbol{W}_{\mathrm{enc}} \in \mathbb{R}^{h \times d}$ and $\boldsymbol{W}_{\mathrm{dec}} \in \mathbb{R}^{d \times h}$ are the encoder and decoder weight matrices, respectively; $\boldsymbol{b}_{\mathrm{enc}} \in \mathbb{R}^h$ and $\boldsymbol{b}_{\mathrm{pre}} \in \mathbb{R}^d$ are bias terms. The function $\sigma^+(\cdot) = \max(0, \cdot)$ denotes the ReLU activation, and $\mathrm{TopK}(\cdot)$ selects the top $k$ largest elements of the input, zeroing out the rest (as in Gao et al. (2024)). As a result, the latent $z_k$ is always a sparse non-negative vector with $k$ active dimensions. This enables direct control over the accuracy–compute trade-off in downstream tasks, particularly under resource-constrained conditions. We formulate the loss function as follows:

$$\mathcal{L}(k) = \left\| f(x) - \widehat{f(x)}_k \right\|_2^2. \quad (4)$$

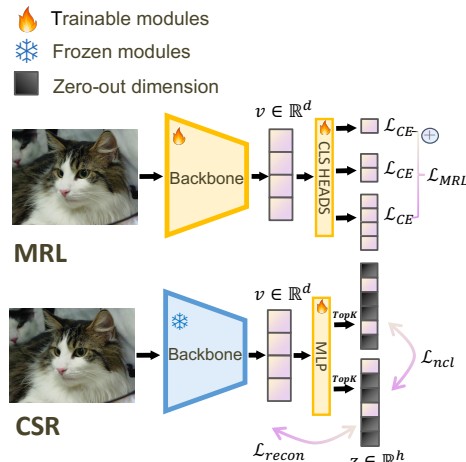

*Figure 2.* Overview of our proposed **CSR** framework. As a post-training approach, CSR differs fundamentally from MRL by projecting embeddings into a higher-dimensional space and dynamically activating only the TopK dimensions for a compact representation. The hidden space is constrained by both reconstruction and contrastive losses, which together enhance the capacity of the sparse representation while preserving computational efficiency.

Moreover, as the hidden dimension $h$ increases, we empirically observe that an increasing number of latent dimensions remain inactive during training – a phenomenon referred to as "dead latents". A large proportion of dead latents reduces the model's capacity and leads to performance degradation (Lu et al., 2019; Templeton et al., 2024). To mitigate this issue, an auxiliary loss $\mathcal{L}_{\mathrm{aux}}$ and Multi-TopK losses are proposed to mitigate this problem. The overall reconstruction loss is

$$\mathcal{L}_{\mathrm{recon}} = \mathcal{L}(k) + \mathcal{L}(4k)/8 + \beta\mathcal{L}_{\mathrm{aux}}, \quad (5)$$

where $\mathcal{L}_{\mathrm{aux}} = \|e - \hat{e}\|_2^2$, $e = f(x) - \widehat{f(x)}$, and $\hat{e} = W_{\mathrm{dec}}z$ is the reconstruction using the top-$k_{\mathrm{aux}}$ dead latents. By default, we set $k_{\mathrm{aux}} = 512$ and $\beta = 1/32$, following the setting in Gao et al. (2024). We also offer dynamic sparsity selection, with $k$ ranging from 8 to 256, to accommodate different tasks across various modalities.

### 3.2.2. SPARSE CONTRASTIVE LEARNING

Furthermore, we consider to incorporate an additional *sparse contrastive loss* to the representations' discriminative power. Most state-of-the-art embedding models today, *e.g.*, CLIP (Radford et al., 2021), follow a contrastive learning paradigm, which that learns to use the embeddings to distinguish between positive and negative pairs. And it applies to both supervised and unsupervised settings (Huang et al., 2024).

The loss objective can be formulated as:

$$\mathcal{L}_{cl} = -\frac{1}{\mathcal{B}} \sum_{i=1}^{\mathcal{B}} \log \frac{exp(z_i^T z_i)}{exp(z_i^T z_i) + \sum_{j \neq i}^{\mathcal{B}} exp(z_i^T z_j)}. \quad (6)$$

By leveraging the non-negative nature of latent variables $z_i$ in sparse autoencoders, Equation 6 can be viewed as a variant of the Non-negative Contrastive Loss (NCL) proposed in Wang et al. (2024). This interpretation enables us to draw on the theoretical guarantees of NCL, as stated in the following theorem:

**Theorem 5** (Wang et al. (2024)). *Under mild conditions, the solution $\phi(x)$ is the unique solution to the NCL objective. As a result, NCL features are identifiable and disentangled.*

Theoretically guaranteed by Theorem 5, the model is encouraged to utilize a larger number of latent dimensions to reconstruct the input data. This behavior is empirically demonstrated in Figure 6, where we observe a reduction in "dead" dimensions compared to vanilla SAE approaches.

### 3.2.3. OVERALL TRAINING OBJECTIVE

At last, we optimize the sparse module through a combination of sparse autoencoding $\mathcal{L}_{recon}$ and sparse contrastive learning $\mathcal{L}_{ncl}$. The former incentivizes the model to preserve original semantic information in the original representation, while the latter shapes the sparse representation to be better at discriminative tasks. The final training objective of our Contrastive Sparse Representation (CSR) method is formulated as:

$$\mathcal{L}_{CSR} = \mathcal{L}_{recon} + \gamma \mathcal{L}_{ncl}. \quad (7)$$

Here, $\gamma$ is a hyperparameter that balances the two loss components and is set to 1 by default.

## 4. Empirical Analysis

In this section, we conduct a careful study on the empirical performance of the proposed CSR. All experiments in this section are conducted on ImageNet, using 1-NN accuracy (Johnson et al., 2019) as the evaluation metric. By default, we set the hidden dimension $h$ of CSR to be $h = 4d$, where $d$ is the dimension of the pretrained dense embeddings, and set the default active dimension to $k = 32$.

For a fair and intuitive comparison of MRL and CSR, First, we adopt the notion of *active dimension* as a surrogate metric to benchmark the retrieval time under dense (MRL-type) and sparse (CSR-type) embeddings. For example, "Active Dim = 8" denotes either a length-8 dense embedding (MRL) or a sparse embedding with TopK ($k = 8$) activation (CSR). Notably, we choose it because dense and sparse matrix multiplication have the same computation complexity under the same active dimension $k$, *i.e.*, $\mathcal{O}(k)$.

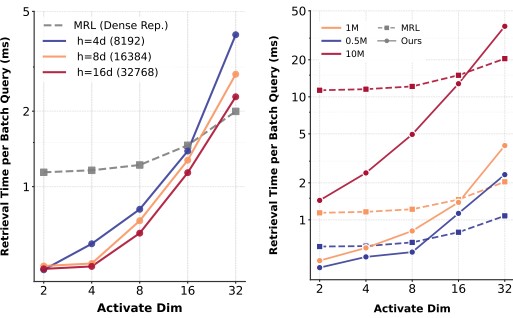

(a) Effect of hidden dim    (b) Effect of database size

*Figure 3.* Comparision of retrieval time based on different factors. (a) Fixed-scale scenario (1M database): Both methods achieve performance sweet spots at TopK=16, with CSR exhibiting 2.1× speedup over dense embeddings when sparsity exceeds 80%. (b) Scaling scenario ($h = 8192$): CSR exhibits increasingly efficient scalability from 0.5M to 10M, with performance gains accelerating at larger scales. This makes it highly practical for real-world applications involving millions of entries.

In Section 4.1, we further carefully benchmark them in practice and find that the two indeed have similar retrieval time, and sparse ones can be even slightly faster under small $k$.

To account for variations in retrieval time due to sample size, we establish a standardized benchmarking protocol (denoted as $\mathcal{T}$) to measure retrieval latency by default. Specifically, to simulate large-scale retrieval scenarios, we report the average retrieval time for 512 queries over an ImageNet-scale database containing 1.3 million entries (equivalent to the size of the ImageNet training set). For CSR, we use a default hidden dimension of $h = 16{,}384$ and an active dimension of $k = 32$. All experiments are conducted in a consistent GPU environment using PyTorch (Paszke et al., 2019). To facilitate comparison, we also report the relative retrieval time of each method by normalizing it against the retrieval time of CSR under the default setup. Additional implementation details can be found in Section E.3.

### 4.1. Retrieval Time Comparison with MRL

In this section, we benchmark the retrieval time of MRL and CSR under the same active dimension $k$ and analyze the impact of hidden dimension $\mathbb{R}^h$, database size $N$ and sparsity $k$.

*(i) Active dimension.* Figure 3(a) shows retrieval time under varying hidden dimensions, with database size fixed. We can see that the retrieval time of CSR (*i.e.*, sparse multiplication) and MRL (*i.e.*, dense multiplication) both grow with large $k$ and remain relatively on the same level. And for smaller $k$, CSR shows a clearer advantage over MRL. Although CSR and MRL have similar theoretical complexity $O(dk)$, their actual runtimes are affected by backend im-

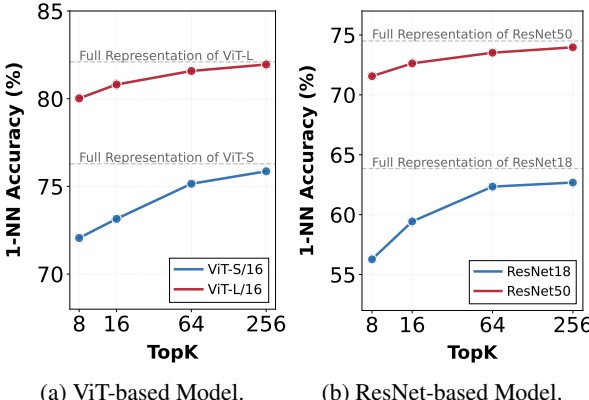

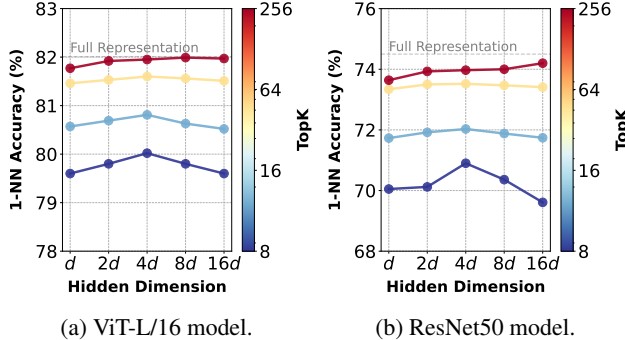

(a) ViT-based Model.  (b) ResNet-based Model.

*Figure 4.* **Performance of CSR under different sparsity levels with different sizes of backbone models.** CSR achieves higher fidelity at greater sparsity levels when applied to larger backbone models (which provide better base performance), observed consistently in both ViT and ResNet architectures.

plementations. For instance, cuBLAS (used for dense ops) is highly optimized but has high launch overhead, while cuSPARSE (used for CSR) is lighter but less optimized for small $k$. Interestingly, we can observe that for sparse embeddings, retrieval time decreases as hidden dimension $h$ increases. This suggests notable benefit of CSR that it can use higher latent dimensions for better expressivity while attaining faster retrieval. On the contrary, MRL with higher dense dimensions always has slower retrieval. We elaborate potential reasons on this distinction at Appendix E.4.

*(ii) Database size.* Figure 3(b) shows that CSR demonstrates superior scalability as the database size $N$ increases from 0.5M to 10M. The relative efficiency gain becomes more pronounced with larger datasets, underscoring the practicality of sparse embeddings in real-world retrieval scenarios.

## 4.2. Effect of Backbone Size

**Experiment Setup.** We examine fidelity versus backbone size (with different input dimension $\mathbb{R}^d$), and sparsity, using fixed hidden dimension $\mathbb{R}^h$ across architectures. For ViT, we use ViT-S/16 ($d = 384$) and ViT-L/16 ($d = 1024$) with $h = 4096$. For ResNet, we test RN18 ($d = 512$) and RN50 ($d = 2048$) with $h = 8192$. A more detailed experiment setup is provided in Section E.1.

**Analysis.** Figure 4 demonstrates that a larger backbone with higher input embedding dimensions improves model fidelity at equal sparsity levels. This insight is particularly significant, as larger embedding sizes generally encode richer information, thereby achieving better downstream performance. By leveraging these high-dimensional embeddings, our approach more effectively retains essential features and relationships within the data.

(a) ViT-L/16 model.  (b) ResNet50 model.

*Figure 5.* **Performance of CSR under different hidden dimensions and different types of backbone models (ResNet-50 (convolution) and ViT-L (Transformers)).** CSR exhibits a reverse U-shape across different models and hidden dimensions. CSR's performance peaks at $h = 4d$ ($d$ is the input dimension size) but degrades beyond this, especially with higher sparsity.

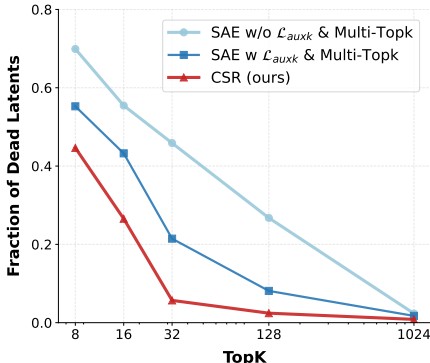

*Figure 6.* Comparison of dead latent fractions across loss combinations under varying sparsity constraints. Results show that even equipped with $\mathcal{L}_{\text{auxk}}$ and Multiple-TopK at extreme sparsity levels (*i.e.*, $k = 8, 16, 32$). CSR further alleviates this issue, outperforming baselines and demonstrating its robustness.

## 4.3. Effect of Hidden Representation Dimension $\mathbb{R}^h$

**Experiment Setup.** We explore how hidden dimension $\mathbb{R}^h$ effects on our model, we use ViT-Large and ResNet50 as pre-trained backbones, sweeping $h$ from $d$ to $16d$ while keeping all other parameters at their default values. Additional implementation details are provided in Section E.2.

**Analysis.** Figure 5 compares model performance across different hidden dimensions under varying sparsity constraints. Notably, a shift in the performance trend occurs at $h = 4d$. When $h < 4d$, performance gradually improves with increasing hidden dimension, reaching its peak at $h = 4d$. However, beyond this point, further increases in $h$ lead to performance degradation, particularly under higher sparsity constraints. This trend aligns with the observations of Gao et al. (2024), which suggest that excessively large

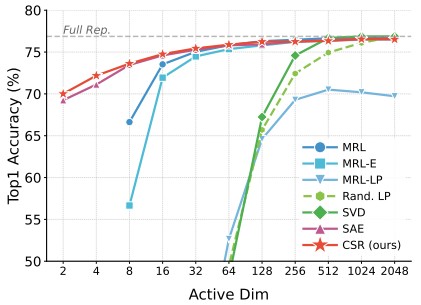

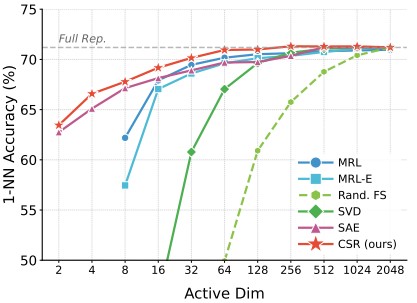

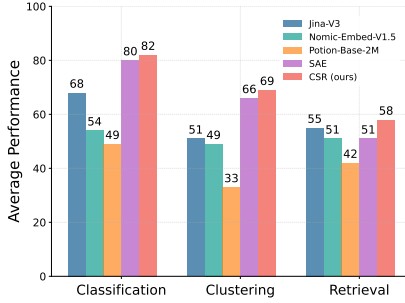

(a) Comparison of ImageNet1k Topk-1 Accuracy under the same pretrained backbone

(b) Comparison of ImageNet1k 1-NN Accuracy under the same pretrained backbone

(c) Comparison of natural language tasks across models with similar retrieval time

*Figure 7.* (**Left & Middle**): Results of ImageNet Top-1 accuracy (a) and 1-NN accuracy (b) across active dimensions under the same pretrained ResNet-50 backbone used in Kusupati et al. (2022). We can see that while MRL trains the whole network and CSR only uses frozen embeddings, CSR still performs consistently better across all embedding sizes and has significant margins beyond 20% at lower active dimensions (the region that yields the largest efficiency gains). (**Right**): *Comparison of text embedding methods at similar retrieval cost.* For CSR, we use $k = 32$ by default. For each task, the model is trained on three datasets and evaluated on three unseen datasets. The text embeddings learned by CSR outperformed other MRL-based baselines by significant margins across different natural language tasks at much lower training cost.

hidden dimensions may not be fully utilized, ultimately diminishing model performance. A similar pattern is observed in ResNet. Based on these findings, we set $h = 4d$ as the default configuration for all subsequent experiments unless otherwise specified.

### 4.4. Effect of Different Losses

**Experiment Setup.** We investigate how different loss functions affect model capacity, particularly in addressing the dead latent problem discussed in Section 3.2.1, using RN50 backbone with $h = 4d$. Other parameters are set at their default values.

**Analysis.** Figure 6 illustrates the impact of different loss functions on model capacity. The naïve SAE suffers from severe dead latents, while the inclusion of an auxiliary loss $\mathcal{L}_{aux}$ and the multi-TopK loss partially mitigates this issue. Introducing a non-negative contrastive loss (NCL) further alleviates the problem, particularly at extreme sparsity levels (*e.g.*, $k = 8, 16, 32$). Empirical results validate the effectiveness of Theorem 5, demonstrating that representation learning with NCL promotes more orthogonal and disentangled features. This, in turn, increases the number of active dimensions and enhances overall model performance.

## 5. Benchmark Results and Analysis

We evaluated the effectiveness of our proposed CSR framework across three mainstream representation modalities: vision, language, and vision+language. For vision representation (see Section 5.1), we conduct image classification on ImageNet-1K and evaluate performance using 1-NN accuracy, following Kusupati et al. (2022). For language

representation (see Section 5.2), we focus on three primary tasks: text classification, text clustering, and text retrieval on the MTEB benchmark (Muennighoff et al., 2022). For multimodal representation (see Section 5.3), we report both in-distribution and zero-shot cross-modal retrieval performance on two widely-used datasets: MS COCO (Lin et al., 2014) and Flickr30K (Young et al., 2014). Through these experiments, we aim to provide a holistic understanding of the capabilities of our proposed framework.

### 5.1. Vision Representation Comparision

**Baselines** We compare our proposed method with the following baseline approaches. 1) MRL/MRL-E (Kusupati et al., 2022): RN50 model where the fully connected layer is replaced by multiple (MRL) or a single (MRL-E) classification head(s) that take truncated input dimensions (*e.g.*, only the first 8 of the original 2048 dimensions). 2) SVD: We performed a low-rank approximation of the 1000-way classification layer of RN50, with rank = 1000. 3) Rand-LP: We compared against a linear classifier fit on randomly selected features (He et al., 2020). 4) Rand-FS: We randomly selected features extracted from RN50 for 1-NN classification.

**Experiment Setup.** We evaluate 1-NN accuracy and Top-1 accuracy on ImageNet1k classification, following Kusupati et al. (2022). For fair comparison, we used the same RN50 backbone weights as MRL (denoted as FF2048 in the original work) and trained CSR on its ImageNet1k encoded embeddings. For further implementation details, please refer to Section B.

*Table 1.* **Performance and efficiency of text embeddings on three natural language tasks: classification, clustering, and retrieval.** We use NV-Embed-V2 as our pre-trained model, and present its performance in the first line of the table in gray. We analyze *Dataset-Specific Evaluation* results along two key dimensions: (1) Relative Retrieval Time under matched performance and ii) performance under matched retrieval efficiency. Under matched performance, CSR achieves a remarkable 61× speedup, while under matched retrieval efficiency, it improves performance by 15%, demonstrating its superior balance between speed and accuracy. The maximum values are indicated in **bold**, while the second-highest values are underlined. Relative Retrieval Time is calculated follows the definition in Section 4.

| Category | Model | Active Dim | Retrieval Time | Text Classification Top-1 Acc (%) ↑ | | | Text Clustering Top-1 Acc (%) ↑ | | | Text Retrieval NDCG@10 (%) ↑ | | |
|---|---|---|---|---|---|---|---|---|---|---|---|---|
| | | | | MTOPIntent | Banking77 | TweetSentiment | BiorxivP2P | BiorxivS2S | TwentyNews | FiQA2018 | NFCorpus | SciFACT |
| Full Rep | NV-Embed-V2 | 4096 | 37.6 | 93.58 | 92.20 | 79.73 | 53.61 | 49.60 | 64.82 | 62.65 | 43.97 | 77.93 |
| MRL | Stella-1.5B-v5 | 256 | 2.6 | **90.45** | 86.14 | 76.75 | 50.81 | 46.42 | 60.07 | 55.59 | 36.97 | **77.48** |
| | Jina-V3 | 256 | 2.8 | 78.81 | 84.08 | 73.81 | 38.14 | 34.39 | 51.96 | 55.73 | 36.63 | 66.63 |
| | Nomic-Embed-V1.5 | 256 | 2.7 | 72.47 | 83.69 | 59.20 | 38.19 | 31.83 | 48.56 | 35.00 | 32.54 | 68.24 |
| | Gecko-Embed-004(Google) | 256 | 2.4 | 77.82 | 86.01 | 72.97 | 36.28 | 33.09 | 50.60 | 55.54 | 37.81 | 70.86 |
| | Text-Embed-3-L (OpenAI) | 256 | 2.8 | 70.45 | 83.19 | 58.98 | 35.43 | 33.86 | 54.24 | 50.33 | 37.94 | 73.10 |
| | Arctic-Embed-L-V2 | 256 | 2.6 | 67.69 | 80.99 | 59.06 | 34.25 | 34.07 | 30.06 | 44.69 | 35.02 | 69.51 |
| | M2V-Base-Glove | 256 | 2.4 | 59.26 | 72.39 | 50.02 | 32.26 | 22.34 | 25.38 | 11.82 | 23.15 | 50.66 |
| | Jina-V3 | 64 | 1.2 | 68.12 | 67.98 | 71.18 | 36.89 | 33.57 | 50.22 | 44.18 | 33.66 | 68.84 |
| | Nomic-Embed-V1.5 | 64 | 1.6 | 62.77 | 80.63 | 55.23 | 34.81 | 44.61 | 48.06 | 10.22 | 18.96 | 36,55 |
| | Potion-Base-2M | 64 | 1.4 | 42.50 | 65.17 | 52.52 | 25.78 | 14.94 | 27.07 | 32.08 | 30.72 | 64.28 |
| Sparse | SAE (w/ NV-Embed-V2) | 32 | 1.0 | 87.43 | 88.11 | 75.19 | 51.02 | 48.68 | 58.63 | 49.18 | 35.14 | 66.04 |
| | **CSR (w/ NV-Embed-V2)** | 32 | 1.0 | 89.86 | **91.02** | **78.55** | **53.49** | **49.13** | **63.05** | **57.54** | **38.06** | 71.17 |

**Analysis.** Figure 7(a) and (b) illustrate the comparison of learned representation quality through the Top-1 and 1-NN classification accuracy of RN50 models trained and evaluated on ImageNet-1K. For linear probing results (Figure 7(a)), reconstruction-based sparse compression methods (CSR & SAE) outperform MRL-LP (both linear probing methods) by a large margin and also surpass MRL/MRL-E (train from scratch) in lower active dim ($k < 128$). Furthermore, Figure 7(a) demonstrates the superior representation quality learned by CSR, which consistently outperforms MRL across various active dimensions. CSR also surpass traditional post-hoc compression techniques (*e.g.*, SVD) and linear probes on random features by increasing the overall model total capacity while keeping active dimensions for each sample unchanged, as discussed in Section 1 and Section 3.2.1. This enhanced capability allows CSR to maintain remarkable robustness, even under extrem sparsity where $k = 2, 4, 8$. These results highlight that the proposed CSR design can effectively compress pre-trained embeddings while leveraging the natural benefits of sparse matrix multiplication. More detailed experimental results can be found in Section 4.

### 5.2. Text Representation Comparision

**Experiment Setup.** We assessed CSR on three key tasks from the MTEB benchmark, testing it across six datasets for each task. In detail, we conduct evaluations in two distinct settings: *Dataset-Specific Evaluation*, where CSR is trained and tested on different splits of the same dataset to ensure consistency, and *Task-Specific Evaluation*, where CSR is trained on one dataset and evaluated on unseen datasets within the same task to rigorously assess its generalization capabilities. We choose NV-Embed-V2 (Lee et al., 2024a) as our pre-trained model and present its performance in gray.

For further experimental details, please refer to Section C. To improve readability, we refer to CSR-K as a model with the TopK activations and so as SAE.

**Analysis** Table 1 demonstrates the performance of CSR and baseline models across multiple tasks and datasets. CSR not only maintains the strong performance of the pre-trained model but also surpasses baselines under varying resource constraints. Taking text classification as an example, CSR achieves a 15% accuracy improvement at matched computational cost (*i.e.*, with retrieval times comparable to Jina-V3-64 and Nomic-Embed-V1.5-64) while attaining a 61x speedup when matched for performance (*i.e.*, compared to NV-Embed-V2). The results underscore CSR 's exceptional ability to maintain an optimal speed-accuracy trade-off - a critical requirement for practical deployment in large-scale retrieval systems. We further evaluate the generalization capability of CSR (with $k = 32$) on three unseen datasets per task, as shown in Figure 7(c). The results demonstrate that sparse representations yield more robust performance compared to dense alternatives at same activation dimensions. These results underscore the efficacy and versatility of CSR , demonstrating its strong potential for real-world applications.

### 5.3. MultiModal Representation Comparision

**Experiment Setup.** We evaluated our methods on multimodal retrieval tasks using the ViT-B-16 backbone, testing both in-distribution and zero-shot cross-modal retrieval on MS COCO (Lin et al., 2014) and Flickr30K (Young et al., 2014) datasets. For baselines, we fine-tuned MRL on these datasets (using CC3M (Changpinyo et al., 2021) for zero-shot training), following standard MRL training protocols (Kusupati et al., 2022). The performance of our backbone,

*Table 2.* **Comparison of different methods on multi-modal retrieval tasks using two benchmark datasets, MS COCO and Flickr30k, evaluated under both in-distribution and zero-shot settings, with Recall@5 (%) as the performance metric.** We use ViT-B/16 as our pre-trained model, and present its performance in the first line of the table in gray. For Zero-Shot setting, CSR is first trained on a large-scale scale, dataset-CC3M and evaluated on downstream tasks. CSR (plug-and-play) consistently outperforms ViT-B-16-MRL (fully fine-tuned) in various tasks with significant training efficiency.

| Method | Active Dim | Trainable Parms | In-Distribution | | | | Zero-Shot | | | |
| | | | MS COCO | | Flickr30K | | MS COCO | | Flickr30K | |
| | | | I2T | T2I | I2T | T2I | I2T | T2I | I2T | T2I |
| ViT-B-16 | 512 | 86M | 74.42 | 86.47 | 91.92 | 97.79 | 69.23 | 83.03 | 89.82 | 97.70 |
| ViT-B-16-MRL | | 86M | 67.12 | 77.53 | 80.41 | 89.89 | 56.90 | 65.82 | 80.94 | 89.20 |
| SAE | 256 | 1.1M | 71.21 | 82.58 | 87.76 | 95.59 | 58.22 | 67.40 | 82.44 | 86.19 |
| **CSR** | | 1.1M | **71.41** | **83.49** | **87.98** | **96.79** | **61.85** | **70.14** | **85.22** | **91.10** |
| ViT-B-16-MRL | | 86M | 64.19 | 73.02 | 77.56 | 87.80 | 53.63 | 61.16 | 77.67 | 85.10 |
| SAE | 128 | 1.1M | 64.67 | 76.70 | 81.40 | 91.20 | 53.20 | 63.02 | 77.54 | 85.19 |
| **CSR** | | 1.1M | **69.34** | **81.04** | **84.05** | **93.00** | **54.37** | **68.04** | **78.08** | **88.09** |
| ViT-B-16-MRL | | 86M | 62.61 | 72.43 | 74.22 | 84.79 | 47.47 | 54.42 | 71.16 | 79.00 |
| SAE | 64 | 1.1M | 56.30 | 69.45 | 70.58 | 81.30 | 44.48 | 53.56 | 69.58 | 82.29 |
| **CSR** | | 1.1M | **62.75** | **78.10** | **76.44** | **88.50** | **48.61** | **61.90** | **73.04** | **84.10** |

using the same fine-tuning procedure, is shown in gray. During training, both SAE and CSR leverage a shared sparse embedding layer for images and text. Additional experimental setup and implementation details are provided in Section D.

**Analysis.** Table 2 presents the multimodal retrieval task results across different methods and settings. In general, reconstruction-based methods exhibit relatively low performance degradation on both datasets. Compared to the MRL method, CSR achieves average performance gains of 4.6% and 6.8% on I2T retrieval, and 10.3% and 6.5% on T2I retrieval across the two datasets in In-Distribution Evaluation. Besides, under zero-shot scenario, CSR also surpasses MRL by 3.2% and 3.3% on I2T, and 9.2% and 3.9% on T2I, respectively. Notably, these results demonstrate CSR's potential to handle large-scale datasets (*e.g.*, CC3M-3M images, compared to ImageNet's 1M and MS COCO's 0.3M), confirming CSR's consistent superiority across various active dimensions and its scalability. SAE experiences more severe performance degradation compared to CSR, which underlines the efficacy of our design in image-text alignment. However, as the sparsity constraint becomes more stringent, the performance gap between CSR and MRL narrows. Upon further investigation, we find that CSR still suffers from the "dead latents" problem even when equipped with advanced mechanisms. Addressing the mitigation of dead latents in the alignment space remains an open challenge, leaving room for future work and study. For a detailed analysis, please refer to Section D.4.

## 6. Conclusion & Discussion

In this paper, we introduce Contrastive Sparse Representation Learning (CSR), a generic learning framework offering a high-fidelity and flexible approach to compress embedding, surpassing existing methods like MRL in various tasks and modalities. We believe CSR paves the way for more efficient and flexible representation learning, especially in scenarios constrained by memory, latency or other computational considerations.

Our method, CSR, is orthogonal to existing acceleration techniques such as pruning (He et al., 2017), quantization (Jacob et al., 2018), and distillation (Hinton et al., 2015), which primarily target embedding generation. In contrast, CSR optimizes the post-processing stage, enabling complementary speedups with minimal performance trade-off. A current limitation of CSR, shared by other sparsity-based approaches, is the emergence of dead neurons under high sparsity, especially in multimodal settings. While techniques like contrastive loss partially mitigate this (see Figure 6), fully resolving the issue remains an open challenge and direction for future work.

## Acknowledgements

This work was supported in part by the National Natural Science Foundation of China under Grant U21B2006; in part by the Fundamental Research Funds for the Central Universities QTZX24003 and QTZX23018; in part by the 111 Project under Grant B18039; and in part by Shaanxi Youth Innovation Team Project.

## Impact Statement

This paper presents work whose goal is to advance the field of Machine Learning. There are many potential societal consequences of our work, none of which we feel must be specifically highlighted here.

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

# A. Datasets

For Image embedding Experiment:

- **ImageNet-1K** (Deng et al., 2009): ImageNet-1K is a large-scale visual database designed to provide researchers with a comprehensive resource for developing and evaluating computer vision models. It contains 1,000 categories, each with a diverse set of images. Specifically, the dataset includes 1,281,167 training images, 50,000 validation images, and 100,000 test images.

For Text embedding Experiment:
Note that, all datasets mentioned below can be found at MTEB (Muennighoff et al., 2022).

- **MTOPIntent** (Li et al., 2020): MTOP is a multilingual dataset introduced in 2021. It comprises 100,000 annotated dialogue sentences across six languages and eleven domains. Designed to serve as a benchmark for multilingual task-oriented semantic parsing, this dataset plays a crucial role in advancing technology in this field.

- **Banking77** (Casanueva et al., 2020): Dataset composed of online banking queries annotated with their corresponding intents, consisting of 13,083 customer service queries labeled with 77 intents.

- **TweetSentimentExtraction** (Maggie et al., 2020): Dataset from Kag gle competition. Sentiment classification of tweets as neutral, positive or negative.

- **MassiveScenario** (FitzGerald et al., 2022): A collection of Amazon Alexa virtual assistant utterances annotated with the associated intent. For each user utterance the label is a theme among 60 scenarios like 'music', 'weather', etc. This is a multilingual dataset with 51 available languages.

- **AmazonReviews** (McAuley & Leskovec, 2013): A collection of Amazonreviews designed to aid research in multilingual text classification. For each review the label is the score given by their view between 0 and 4 (1-5 stars). This is a multilingual dataset with 6 available languages.

- **Emotion** (Saravia et al., 2018): The dataset consists of English Twitter messages categorized into basic emotions, including anger, fear, joy, love, sadness, and surprise.

- **ArxivClusteringS2S, BiorxivClusteringS2S, BiorxivClusteringP2P** (Muennighoff et al., 2022): The BioxivS2S dataset is created using public APIs from bioRxiv. For S2S datasets, the input text is simply the title of the paper, while for P2P the input text is the concatenation of the title and the abstract.

- **TwentyNewsgroupsClustering**[2]: Clustering of the 20 Newsgroups dataset, given titles of article the goal is to find the newsgroup (20 in total). Contains 10 splits, each with 20 classes, with each split containing between 1,000 and 10,000 titles.

- **RedditClusteringP2P** (Muennighoff et al., 2022): created for MTEB using available data from Reddit posts[3]. The task consists of clustering the concatenation of title+post according to their subreddit. It contains 10 splits, with 10 and 100 clusters per split and 1,000 to 100,000 posts.

- **StackExchangeClustering** (Geigle et al., 2021): Clustering of titles from 121 stack exchanges. Clustering of 25 splits, each with 10-50 classes, and each class with 100-1000 sentences.

- **FiQA2018** (Maia et al., 2018): A dataset for aspect-based sentiment analysis and opinion-based question answering in finance.

- **NFCorpus** (Boteva et al., 2016): NFCorpus is a full-text English retrieval data set for Medical Information Retrieval. It contains a total of 3,244 natural language queries, with 169,756 automatically extracted relevance judgments for 9,964 medical documents.

- **SciFACT** (Wadden et al., 2020): A dataset of 1.4K expert-written claims, paired with evidence-containing abstracts annotated with veracity labels and rationales.

---

[2]https://scikit-learn.org/0.19/datasets/twenty_newsgroups.html
[3]https://huggingface.co/datasets/sentence-transformers/reddit-title-body

- **Arguana** (Wachsmuth et al., 2018b): The dataset consists of debates from idebate.org, collected as of January 30, 2018. Each debate includes the thesis, introductory text, all points and counters, bibliography, and metadata.

- **CQADupStack** (Hoogeveen et al., 2015): A benchmark dataset for community question-answering research. It contains threads from twelve StackExchange subforums, annotated with duplicate question information.

- **Quora Question Pairs**[4]: A dataset consists of over 400,000 question pairs, and each question pair is annotated with a binary value indicating whether the two questions are paraphrase of each other.

For Multimodal embedding Experiment:

- **MS COCO** (Lin et al., 2014): The MS COCO dataset is a large-scale object detection, segmentation, and captioning dataset. It contains images with complex scenes involving multiple objects, each annotated with labels, bounding boxes, and segmentation masks.

- **Flickr30K** (Young et al., 2014): The Flickr30k dataset is a collection of images with corresponding textual descriptions. Each image is annotated with multiple captions that describe the scene, objects, and actions depicted.

## B. Experiment Detail on Vision Representation.

### B.1. Evaluation Metric

We adopt 1-NN as our evaluation metric, implemented using FAISS (Johnson et al., 2019) with exact L2 search, following the setup in (Kusupati et al., 2022). This approach provides an efficient and cost-effective way to evaluate the utility of learned representations for downstream tasks, as 1-NN accuracy requires no additional training. In detail, we use the training set with 1.3M samples as the database and the validation set with 50K samples as the query set. We also report linear probing and few-shot results using Top-1 accuracy. For a holistic evaluation, different methods, Figure 1 (c) presents the average 1-NN performance (active dimensions $< 64$).

### B.2. Baselines

We select MRL and MRL-E from (Kusupati et al., 2022) as baselines. This work introduces a novel training paradigm that learns representations of varying lengths. MRL-E is an efficient version of MRL, also proposed in (Kusupati et al., 2022).

### B.3. Implementation Detail

For a fair comparison, we selected the pre-trained ResNet50 weights, noted as FF2048 in the MRL (Kusupati et al., 2022). Additionaly, we select the ResNet50 model[5] as our SOTA backbone from Wightman (2019). For image preprocessing, we adopt the same procedure as described in Kusupati et al. (2022); Leclerc et al. (2023). Consistent with Gao et al. (2024), we utilize a tied encoder-decoder structure to build the CSR framework. The implementation of CSR is based on the codebase[6] provided by OpenAI. All experiments are conducted on a server equipped with 4 RTX4090 GPUs. The selection of hyperparameters are:

*Table 3.* Implementation details on Image experiment.

| Backbone | $d$ | $h$ | lr | epoch | Batch Size | $k_{\text{aux}}$ | $\beta$ | $\gamma$ | $\mathbb{K}$ | Optimizer | weight decay | eps |
|---|---|---|---|---|---|---|---|---|---|---|---|---|
| ResNet50 | 2048 | 8192 | 4e-5 | 10 | 4096 | 512 | 1/32 | 0.1 | 8,16,32...2048 | Adam | 1e-4 | 6.25 * 1e-10 |

### B.4. 1-NN Classification Results

1-NN classification and Top-1 linear probing results are shown in Table 4 and Table 5.

---

[4] https://paperswithcode.com/dataset/quora-question-pairs
[5] https://huggingface.co/timm/resnet50d.ra4_e3600_r224_in1k
[6] https://github.com/openai/sparse_autoencoder

*Table 4.* 1-NN accuracy of different methods on ImageNet1k classification.

| Active Dim | Full Rep. | MRL | MRL-E | SVD | Rand. FS | SAE | CSR | SOTA Full Rep. | CSR (w/ SOTA RN50) |
|---|---|---|---|---|---|---|---|---|---|
| 8 | - | 62.19 | 57.45 | 19.14 | 2.36 | 67.14 | 67.78 | - | 73.84 |
| 16 | - | 67.91 | 67.05 | 46.02 | 12.06 | 68.14 | 69.17 | - | 74.39 |
| 32 | - | 69.46 | 68.60 | 60.78 | 32.91 | 68.91 | 70.15 | - | 74.53 |
| 64 | - | 70.17 | 69.61 | 67.04 | 49.91 | 69.69 | 70.94 | - | 74.62 |
| 128 | - | 70.52 | 70.12 | 69.63 | 60.91 | 69.74 | 70.99 | - | 74.65 |
| 256 | - | 70.62 | 70.36 | 70.67 | 65.75 | 70.35 | 71.31 | - | 74.73 |
| 512 | - | 70.82 | 70.74 | 71.06 | 68.77 | 71.21 | 71.29 | - | 74.88 |
| 1024 | - | 70.89 | 71.07 | 71.22 | 70.41 | 71.20 | 71.30 | - | 74.90 |
| 2048 | 71.19 | 70.97 | 71.21 | 71.21 | 71.19 | 71.24 | 71.20 | 75.19 | 74.91 |

*Table 5.* Top-1 classification accuracy results of different methods on ImageNet1k classification.

| Active Dim | Full Rep. | MRL | MRL-E | MRL-LP | SVD | Rand. LP | SAE | CSR | SOTA Full Rep. | CSR (w/ SOTA RN50) |
|---|---|---|---|---|---|---|---|---|---|---|
| 8 | - | 66.63 | 56.66 | 5.15 | 2.34 | 4.56 | 73.46 | 73.62 | - | 79.17 |
| 16 | - | 73.53 | 71.94 | 13.79 | 7.17 | 11.29 | 74.60 | 74.75 | - | 79.72 |
| 32 | - | 75.03 | 74.48 | 32.52 | 20.46 | 27.21 | 75.28 | 75.44 | - | 79.96 |
| 64 | - | 75.82 | 75.35 | 52.66 | 48.10 | 49.47 | 75.81 | 75.88 | - | 80.16 |
| 128 | - | 76.30 | 75.80 | 64.60 | 67.24 | 65.70 | 75.91 | 76.24 | - | 80.24 |
| 256 | - | 76.47 | 76.22 | 69.29 | 74.59 | 72.43 | 76.27 | 76.25 | - | 80.31 |
| 512 | - | 76.65 | 76.36 | 70.51 | 76.78 | 74.94 | 76.43 | 76.34 | - | 80.33 |
| 1024 | - | 76.76 | 76.48 | 70.19 | 76.87 | 76.10 | 76.59 | 76.54 | - | 80.32 |
| 2048 | 76.87 | 76.80 | 76.51 | 69.72 | - | 76.87 | 76.66 | 76.52 | 80.59 | 80.35 |

## C. Experiment Detail on Text Representation

### C.1. Evaluation Metric

We adopt the universal evaluation metrics used in the MTEB benchmark (Muennighoff et al., 2022). For text classification and clustering, we use Top-1 accuracy to assess model performance. For the text retrieval task, we use NDCG@10 (Normalized Discounted Cumulative Gain at 10), a metric that evaluates the quality of a ranked list of items, commonly used in information retrieval and recommendation systems.

### C.2. Experiment Setup

We choose three main tasks on MTEB benchmark and randomly select six datasets(for each task) to measure our methods. We also design two experiment settings to evaluate the effectiveness and generalization ability of our methods.

Firstly, we introduce *Dataset-Specific Evaluation*, where CSR are trained and tested on different splits of the same dataset. We use MTOPIntent (Li et al., 2020), Banking77 (Casanueva et al., 2020) and TweetSentimentExtraction (Maggie et al., 2020) for text classification task. We use BiorxivClusteringS2S, BiorxivClusteringP2P (Muennighoff et al., 2022) and TwentyNewsgroupdClustering for text clustering. For text retrieval, we select FiQA2018 (Maia et al., 2018), NFCorpus (Boteva et al., 2016) and SciFACT (Wadden et al., 2020).

Furthermore, we introduce *Task-Specific Evaluation*, where CSR are trained and tested on different datasets within the same task to evaluate the generalization ability of our proposed method. We construct a training dataset using the training splits of the aforementioned datasets and test on the corresponding task datasets. For classification: MassivScenario (FitzGerald et al., 2022), AmazonRevies (McAuley & Leskovec, 2013) and Emotion (Saravia et al., 2018). For clustering: ArxivClusteringS2S, RedditClusteringP2P (Muennighoff et al., 2022) and StackExchangeClustering (Geigle et al., 2021). For retrieval: Arguana (Wachsmuth et al., 2018a), CQADupStack (Hoogeveen et al., 2015) and Quora.

### C.3. Baselines

We choose several models that provide MRL embeddings on MTEB benchmark (Muennighoff et al., 2022). These models are Stella-en-1.5B-v5 (Zhang et al., 2025), Jina-V3 (Sturua et al., 2024), Nomic-Embed-V1.5 (Nussbaum et al., 2024), Gecko-Text-Embedding-004-256 (Lee et al., 2024b), OpenAI-Text-Embedding-3-L-256 (OpenAI, 2024), Arctic-Embed-L-V2.0 (Yu

et al., 2024) and Potion-Base-2M (min, 2024).

### C.4. Implementation Detail

We select NV-Embed-V2 (Lee et al., 2024a) as our pre-trained model. We utilize a tied encoder-decoder structure to build the CSR framework. For text classification and clustering tasks, we use data from the same class as positive samples while the other as negative samples to calculate Equation 6. The hyperparameters are set as follows:

*Table 6.* Implementation details on Text experiment.

| Backbone | $d$ | $h$ | lr | epoch | Batch Size | $k_{aux}$ | $\beta$ | $\gamma$ | $\mathbb{K}$ | Optimizer | weight decay | eps |
|---|---|---|---|---|---|---|---|---|---|---|---|---|
| NV-Embed-V2 | 4096 | 16384 | 4e-5 | 10 | 128 | 1024 | 0.1 | 1.0 | 32,64,256 | Adam | 1e-4 | 6.25 * 1e-10 |

## D. Experiment Detail on MultiModal Representation

### D.1. Evaluation Metric

We adopt the universal evaluation metric Recall@5 to measure performance in the MultiModal Retrieval task. This metric evaluates a model's ability to retrieve relevant items within its top 5 predictions. Calculated as the fraction of relevant items appearing in the top 5 results out of the total relevant items, a higher Recall@5 indicates better performance in capturing relevant content early in the ranked list, making it useful for recommendation systems and retrieval tasks.

### D.2. Experiment Setup

We selected ViT-B-16, trained on the DFN2B dataset[7], as our pre-trained model. For the in-distribution cross-modal retrieval experiment, we implemented MRL in the pre-trained ViT model following Kusupati et al. (2022), and fine-tuned it for 50 epochs on the MSCOCO (Lin et al., 2014) and Flickr30K (Young et al., 2014) datasets, respectively. For a fair comparison, we also fine-tuned the backbone on both datasets for 50 epochs using the same hyperparameters, which were then used for the backbone of CSR . The hyperparameters used for fine-tuning are as follows:

*Table 7.* Hyperparameters for fine-tuning ViT-B/16 backbone.

| Dataset | lr | epoch | Batch Size | warmup | Optimizer | weight decay |
|---|---|---|---|---|---|---|
| MS COCO | 5e-6 | 50 | 64 | 10000 | Adam | 0.1 |
| Flickr30k | 5e-6 | 50 | 64 | 10000 | Adam | 0.1 |

For zero-shot cross-modal retrieval, we employed the same MRL fine-tuning procedure as in our in-distribution experiment, maintaining identical hyperparameters while training for 3 epochs with 2208 batch size on CC3M (Changpinyo et al., 2021).

### D.3. Implementation Detail

We select the ViT-B-16[8] as our backbone from Wightman (2019). Consistent with Gao et al. (2024), we utilize a tied encoder-decoder structure to build the CSR framework. The encoder and decoder structure share between image space and text space. The implementation of CSR is based on the codebase[9] and OpenCLIP (Cherti et al., 2023). The metric is evaluated through CLIP-benchmark following standard procedure. All experiments are conducted on a server equipped with 4 RTX4090 GPUs. We present detailed training parameters for the multimodal experiment in Table 8.

### D.4. Discussion On Dead Latents

Addressing the mitigation of dead latents in the alignment space remains an open challenge, leaving room for future work and study. Table 2 presents the performance comparison between CSR and MRL, revealing that the gap between the two

---

[7] https://huggingface.co/apple/DFN2B-CLIP-ViT-B-16
[8] https://huggingface.co/apple/DFN2B-CLIP-ViT-B-16
[9] https://github.com/openai/sparse_autoencoder

*Table 8.* Implementation details on MultiModal experiment.

| Dataset | $d$ | $h$ | lr | epoch | Batch Size | $k_{\text{aux}}$ | $\beta$ | $\gamma$ | $\mathbb{K}$ | Optimizer | weight decay | eps |
|---------|-----|-----|-----|-------|------------|-------|---------|----------|--------------|-----------|--------------|-----|
| MS COCO | 512 | 2048 | 4e-4 | 5 | 256 | 512 | 1/32 | 1.0 | 64,128,256 | Adam | 1e-4 | 6.25 * 1e-10 |
| Flickr30k | 512 | 2048 | 4e-4 | 5 | 64 | 1024 | 1.0 | 1.0 | 64,128,256 | Adam | 1e-4 | 6.25 * 1e-10 |
| CC3M | 512 | 4096 | 4e-4 | 1 | 1024 | 1024 | 1/32 | 1.0 | 64,128,256 | Adam | 0.0 | 6.25 * 1e-10 |

methods diminishes as sparsity constraints become more stringent. Further analysis indicates that CSR continues to face the "dead latents" issue despite incorporating advanced mechanisms. As shown in Figure 8, CSR exhibits a significant performance drop, corresponding to a sharp rise in dead latent dimensions. We attribute this to a technical challenge, as CSR has demonstrated robust performance in both image and text domains under similar sparsity constraints. This suggests that representations in alignment spaces may require more specialized design, presenting an opportunity for future research and improvement.

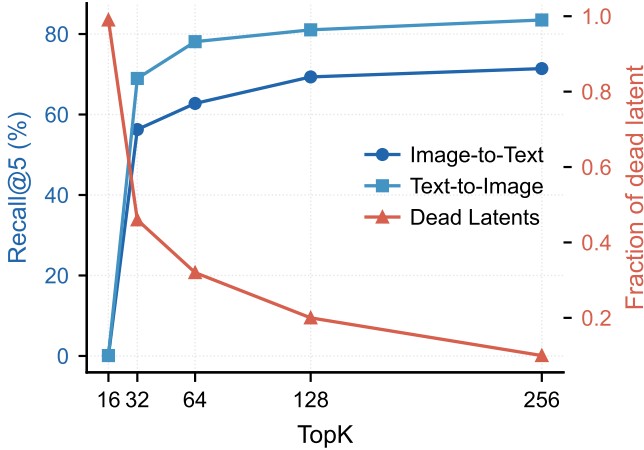

*Figure 8.* Dead latents still exits in image-text alignment space.

# E. Empirical Analysis

### E.1. Effect on Input Embedding Dimension $\mathbb{R}^d$

The implementation details are shown in Table 9. To avoid other unknown factors, we choose ViT-based[10] and ResNet-based models[11] following same pre-training procedure respectively. To ensure generalizability, we train the model using three different random seeds and report the mean performance in the main paper.

*Table 9.* Implementation details on empirical study of input embedding dimension $\mathbb{R}^d$

| Backbone | $d$ | $h$ | lr | epoch | Batch Size | $k_{\text{aux}}$ | $\beta$ | $\gamma$ | $\mathbb{K}$ | Optimizer | weight decay | eps |
|----------|-----|-----|-----|-------|------------|-------|---------|----------|--------------|-----------|--------------|-----|
| ViT-L/16 | 512 | 4096 | 4e-5 | 10 | 1024 | 512 | 1/32 | 1.0 | 8,16,64,256 | Adam | 1e-4 | 6.25 * 1e-10 |
| ViT-L/16 | 1024 | 4096 | 4e-5 | 10 | 1024 | 512 | 1/32 | 1.0 | 8,16,64,256 | Adam | 1e-4 | 6.25 * 1e-10 |
| ResNet18 | 512 | 8192 | 4e-5 | 10 | 1024 | 512 | 1/32 | 1.0 | 8,16,64,256 | Adam | 1e-4 | 6.25 * 1e-10 |
| ResNet50 | 2048 | 8192 | 4e-5 | 10 | 1024 | 512 | 1/32 | 1.0 | 8,16,64,256 | Adam | 1e-4 | 6.25 * 1e-10 |

---

[10] https://huggingface.co/timm/vit_small_patch16_224.augreg_in21k_ft_in1k,https://huggingface.co/timm/vit_large_patch16_224.augreg_in21k_ft_in1k
[11] https://huggingface.co/timm/resnet18.a1_in1k,https://huggingface.co/timm/resnet50.a1_in1k

Table 10. Implementation details on empirical study of hidden dimension $\mathbb{R}^h$

| Backbone | $d$ | $h$ | lr | epoch | Batch Size | $k_{\text{aux}}$ | $\beta$ | $\gamma$ | $\mathbb{K}$ | Optimizer | weight decay | eps |
|---|---|---|---|---|---|---|---|---|---|---|---|---|
| | 1024 | 1024 | 4e-5 | 10 | 1024 | 512 | 1/32 | 1.0 | 8,16,64,256 | Adam | 1e-4 | 6.25 * 1e-10 |
| | 1024 | 2048 | 4e-5 | 10 | 1024 | 512 | 1/32 | 1.0 | 8,16,64,256 | Adam | 1e-4 | 6.25 * 1e-10 |
| ViT-L/16 | 1024 | 4096 | 4e-5 | 10 | 1024 | 512 | 1/32 | 1.0 | 8,16,64,256 | Adam | 1e-4 | 6.25 * 1e-10 |
| | 1024 | 8192 | 4e-5 | 10 | 1024 | 512 | 1/32 | 1.0 | 8,16,64,256 | Adam | 1e-4 | 6.25 * 1e-10 |
| | 1024 | 16384 | 4e-5 | 10 | 1024 | 512 | 1/32 | 1.0 | 8,16,64,256 | Adam | 1e-4 | 6.25 * 1e-10 |
| | 2048 | 2048 | 4e-5 | 10 | 1024 | 512 | 1/32 | 1.0 | 8,16,64,256 | Adam | 1e-4 | 6.25 * 1e-10 |
| | 2048 | 4096 | 4e-5 | 10 | 1024 | 512 | 1/32 | 1.0 | 8,16,64,256 | Adam | 1e-4 | 6.25 * 1e-10 |
| ResNet50 | 2048 | 8192 | 4e-5 | 10 | 1024 | 512 | 1/32 | 1.0 | 8,16,64,256 | Adam | 1e-4 | 6.25 * 1e-10 |
| | 2048 | 16384 | 4e-5 | 10 | 1024 | 512 | 1/32 | 1.0 | 8,16,64,256 | Adam | 1e-4 | 6.25 * 1e-10 |
| | 2048 | 32768 | 4e-5 | 10 | 1024 | 512 | 1/32 | 1.0 | 8,16,64,256 | Adam | 1e-4 | 6.25 * 1e-10 |

### E.2. Effect on Hidden Representation Dimension $\mathbb{R}^h$

Implementation details are shown in Table 10. The pre-trained ViT-L/16[12] and ResNet50 models[13] can be found at timm (Wightman, 2019). To ensure generalizability, we train the model using three different random seeds and report the mean performance in the main paper.

### E.3. Retrieval Time Evaluation

We employ PyTorch (Paszke et al., 2019) to measure retrieval time on ImageNet1k. The average retrieval time is computed over 2000 rounds with a batch size of 512 queries, excluding an initial 100 warm-up rounds. For the learned CSR representation, both query and key embeddings are stored in csr format, and sparse product operations are utilized for similarity computation while maintaining identical experimental settings for fair comparison.

### E.4. Understanding Retrieval Time Difference between Dense and Sparse Embeddings

Although CSR and MRL have similar theoretical complexity $O(k)$, their actual runtimes are affected by backend implementations. For instance, cuBLAS (used for dense ops) is highly optimized but has high launch overhead, while cuSPARSE (used for CSR) is lighter but less optimized for small $k$. Here, we can share a preliminary insight into why sparse embeddings can be faster than dense embeddings and why it can get faster with larger hidden dimension $h$.

Sparse matrix multiplication benefits from zero-skipping: only overlapping non-zero entries are used. For each query, computing the $i$-th output only involves comparing indices of non-zero entries—an integer operation much cheaper than floating-point multiplication. As $h$ increases and $k$ stays small, overlap likelihood drops, reducing the number of multiplications required. In Table We empirically verify this by counting the number of multiplications under various $h$:

Table 11. Comparison on the number of multiplication operation between MRL (dense) and CSR (embeddings) on the default setup.

| Active Dim | MRL | CSR ($h = 8192$) | CSR ($h = 16384$) | CSR ($h = 32768$) |
|---|---|---|---|---|
| 2 | $1.3 \times 10^9$ | $3.2 \times 10^5$ | $1.7 \times 10^5$ | $8.4 \times 10^4$ |
| 4 | $2.6 \times 10^9$ | $1.3 \times 10^6$ | $6.7 \times 10^5$ | $3.4 \times 10^5$ |

The number of operations in CSR is several orders of magnitude smaller than in MRL, and it decreases with larger $h$. This counterintuitive yet practical effect highlights the appeal of using sparse high-dimensional embeddings: they allow richer representations while improving runtime.

---

[12] https://huggingface.co/timm/vit_large_patch16_224.augreg_in21k_ft_in1k
[13] https://huggingface.co/timm/resnet50.a1_in1k

