# OpenReview forum: "Beyond Matryoshka: Revisiting Sparse Coding for Adaptive Representation"
_ICML.cc/2025/Conference — ICML 2025 oral_

### Official Review · Reviewer_BiSs · 2025-03-10

**Overall Recommendation:** 5

**Summary:**

This paper presents a novel approach to the problem of learning adaptive-length representations. While previous methods, particularly MRL, have shown good performance, this work carefully studies the utility of high-dimensional but sparse representations, as opposed to lower dimensional but dense representation, for the adaptive-length setting. To accomplish this, the authors use sparse autoencoders and introduce a contrastive-based loss for training. Their method has significantly reduced training time, and the authors claim better performance over MRL and relevant baselines.

**Claims And Evidence:**

The claims are clear, although I have concerns on the experimental evidence (see my concerns on experimental designs or analyses below).

**Essential References Not Discussed:**

None

**Experimental Designs Or Analyses:**

The experiments are well-designed, and most are sound. The experiment that requires careful attention is the one proposed in Section 4.1 and shown in Figure 3. Namely, I'm not convinced of this paper's timing results, and the authors lack a clear explanation as to why their method is so much faster than MRL. This is especially clear from the language; around line 246 in the second column, the authors say that the decrease in retrieval time is $\textit{likely}$ due to efficient sparse matrix multiplications, and that their results $\textit{suggest}$ that "higher sparsity enables more effective utilization of sparse matrix operations."

To explain why I'm skeptical, a >2x speedup is achieved when the number of active dimensions is 2. In this case, the comparison in speed is between a length-2 dense representation and a length-(8192,16384,32768) representation with only 2 active dimensions. According to Section E.3, csr format is used along with sparse matrix operations. However, there is still an overhead associated with using csr format, and, considering that both CSR and MRL have only two active dimensions in this case, I find it very counterintuitive that CSR can be over 2x faster.

I've noted that the authors reported the normalized retrieval time, and I'm wondering if this is may be a reason for this discrepancy. The authors introduced a base-time metric $\mathcal{T}$, the utility of which is unclear to me. How $\mathcal{T}$ is recorded is also unclear to me. It sounds like the authors used CSR with $h=16384, k=32$, and then normalized all timings by this value? First, I hope the authors can clarify if my understanding is correct or not. Second, I hope the authors can elaborate on why $\mathcal{T}$ is needed. They say "This metric enables a more realistic simulation of large-scale retrieval scenarios for fair computation comparison." Still, they do not elaborate on why normalizing $\mathcal{T}$ makes the comparison more fair.

Indeed, the timing results are somewhat plausible, but I hope the authors can clarify how this speedup is achieved. Is it strictly a result of sparse matrix operations as opposed to some architecture changes over MRL? How does the base time metric affect the timings over the raw timings?

**Methods And Evaluation Criteria:**

Proposed methods and evaluation criteria are relevant

**Other Comments Or Suggestions:**

Some typos that I found:
- first column, line 69: "more fast" --> "faster"
- second column, line 272: this sentence seems incomplete/incorrect
- first column, line 297: "from high-dimensional to high-dimensional" --> "from high-dimensional to low-dimensional" (?)

**Other Strengths And Weaknesses:**

Some strengths include:
- Clear presentation of ideas, experiments, and results
- Paper is written well, aside from a few typos
- Thoughtful and extensive experiment design

**Questions For Authors:**

Please see the questions in the Experimental Designs section. Answering these questions is important and may warrant additional clarification in the paper. I'll repeat the questions here to make them available in a list format:
1. How $\mathcal{T}$ is measured is also unclear to me. It sounds like the authors used CSR with $h=16384, k=32$, and then normalized all timings by this value?
2. Why is $\mathcal{T}$ needed? Why does it make the timing comparisons more fair?
3. How does the base time metric affect the timings over the raw timings?
4. Are timings improvements strictly a result of sparse matrix operations as opposed to some architecture changes over MRL? Is there anything the authors don't currently consider that may explain the speedup?

**Relation To Broader Scientific Literature:**

This work presents an interesting and useful alternative to MRL, the leading method for length-adaptive representations. The use of sparse autoencoders in this setting is novel.

**Theoretical Claims:**

N/A

---

> ### Author Rebuttal · Authors · 2025-04-01
>
> Thanks for your detailed reading and valuable comments. We will address your concerns as follows.
>
> ---
>
> **Q1** Typos in line 69,272 and 297
>
> **A1.** Thanks for your suggestions and we will fix the typos in the revision.
>
> ---
> **Q2** The experiment that requires careful attention is the one proposed in Section 4.1 and shown in Figure 3. Namely, I'm not convinced of this paper's timing results, and the authors lack a clear explanation as to why their method is so much faster than MRL.
> > To explain why I'm skeptical, a >2x speedup is achieved when the number of active dimensions is 2. In this case, the comparison in speed is between a length-2 dense representation and a length-(8192,16384,32768) representation with only 2 active dimensions.
>
> **A2.** We believe there may be some misunderstandings, and we’d like to clarify a key point.
> Most importantly, the speedup shown in Figure 3 (comparing sparse over dense computation) is **not the main reason** for CSR’s efficiency gains.
> The main goal of Figure 3 is to simply show that sparse MM can have similar (sometimes faster) computation to dense MM under the same active dimension (x-axis).
> The main gain of MRL is that it can use a smaller active dimension to better preserve the model accuracy.
> As shown in the table below (quoted from Table 4 (in Sec B.4.)), CSR only degrades 1.8\% accuracy with 8 active dimensions under MRL degrades 12.4\% accuracy. Therefore, to attain the same accuracy level, CSR models can utilize a **much smaller active dimension**.
> As a result, **even if the ``retrieval time per dim`` is similar** among sparse and dense MM, CSR can still attain significant speedup. And this part is the main reason for CSR's significant improvement in efficiency as we demonstrated in Figure 1(b).
>
> *Table 1: 1-NN performance comparison between MRL and CSR across various activation dim*
>
> | Active Dim| MRL   | CSR   |
> |-|-|-|
> |2|-| 66.17 |
> | 4 | -| 69.97 |
> | 8 | 62.19 | 73.84 |
> |16| 67.91 | 74.39 |
> | 32| 69.46 | 74.53 |
> | 2048 (Full rep with ResNet50) | 70.97 | 75.19 |
>
> ---
> **Q3** Why $\mathcal{T}$ is needed?
>
> **A3.** The datasets used in our paper vary by orders of magnitude. For example, FiQA-2018 has 57,638 entries, while ImageNet-1k contains 1.3M in its database. Considering this, we give an ablation study on database size $N$ in Figure 3(c). Our study demonstrates that the efficiency advantage of sparse methods scales with database size(e.g., 1M, 10M entries), making them ideal for real-world applications. Thus,
> to simulate real-world scenarios (e.g., massive-scale entities like in RAG),  we standardize all datasets to ImageNet-1k's scale (1.3M entries) and use a time-based counter $\mathcal{T}$ to eliminate the effect of $N$ across different datasets.
>
> ---
> **Q4** How $\mathcal{T}$ is measured? It sounds like the authors used CSR with
> $h$=16384,$k$=32, and then normalized all timings by this value?
>
> **A4.** Indeed, we calculated the average time $\mathcal{T}$ for performing 2000 matrix multiplications of two sparse matrices using ``torch.sparse.mm()``. Here is a more detailed breakdown of the evaluation protocol for the retrieval time:
> 1. We precompute the embeddings of all ImageNet training data and store them in standard CSR (compressed sparse row) on GPU memory as the database for retrieval.
> 2. We compute the retrieval time as the average over 2,000 rounds of retrieval, after a warm-up period of 100 rounds. Each time, the query consists of 512 samples randomly drawn from the database,
> The warm-up procedure is common in measuring GPU computation time, cause it eliminates initialization bias.
>
> ---
> **Q5** How does the base time metric affect the timings over the raw timings?
>
> **A5.** It does not affect raw timings, as the relative retrieval time is computed by $(\text{raw timing})/\mathcal{T}$, as shown in Figure 3 and Table 1. We also present the raw timing comparison for MRL in Figure 1(b) to demonstrate our improvements. We will revise a more detailed explanation of $\mathcal{T}$ calculation in Sec. 4 and E.3 to clarify our motivation.
>
> ---
> **Q6** Are timings improvements strictly a result of sparse matrix operations as opposed to some architecture changes over MRL? Is there anything the authors don't currently consider that may explain the speedup?
>
> **A6.** As detailed in A2, we would like to reiterate that the core reason CSR is more efficient than MRL lies in its ability to maintain high fidelity even with very small active dimensions. The primary advantage of CSR is not simply that sparse matrix multiplication outperforms dense computation (as shown in Figure 3), but rather that CSR can achieve comparable model accuracy with significantly fewer active dimensions.
>
> ---
>
> Thank you for your thoughtful questions. We hope the responses provided adequately address your concerns. Please don’t hesitate to reach out if any further clarification is needed.

---

> > ### Comment · Reviewer_BiSs · 2025-04-03
> >
> > Thanks for your response and clarifications, especially on the role of $\mathcal{T}$.
> >
> > I acknowledge that Figure 3a is not the main result for the efficiency gains. CSR achieves much better accuracy over MRL for the same number of active dimensions. However, the claim of Figure 3a is that extremely sparse but high-dimensional matrix multiplies are on average faster than dense but low-dimensional matrix multiplies. Again, it is very surprising to me that the speedups are >2x when the active dimension is 2. At the very least, I would expect that there is a crossing point between the MRL and CSR lines such that MRL is faster with respect to the experiment done for Figure 3a. By inspection, this might occur when $k=32$, as the growth rate of the normalized retrieval time for CSR is faster than MRL.
> >
> > I think this is a good paper overall, and will therefore keep my score for now. But I would happy to raise the score if there is a more definitive explanation why being more sparse improves the timings. According to Figure 3a, when $k \geq 4$, higher sparsity leads to even better timings for a fixed active dimension. What I want to ask the authors is the following about Figure 3a: Is the discrepancy among the timings (between MRL and CSR, and then also the timing improvement of CSR as $h$ changes) purely a result of efficient sparse matrix operations in PyTorch, or is it a difference in MRL vs CSR, or a difference in the way the timings were recorded for each method?

---

> > > ### Author Response · Authors · 2025-04-06
> > >
> > > Thank you for your prompt response and clarifying your remaining concerns! We are happy to address them point by point below.
> > >
> > > ---
> > > **Q1.** I expect a crossing point between MRL and CSR performance curves in Figure 3a, potentially at k=32, as CSR's normalized retrieval time appears to increase more rapidly than MRL's.
> > >
> > > **A1.** Indeed, there is a crossover point around $k=32$, where dense retrieval begins to be slightly faster than sparse retrieval (eg 0.0019s vs 0.0022s when $k=32$ and 0.0029s vs 0.0036s when $k=64$) -- but overall, it is on the same scale. This speed difference is not related to CSR or MRL methods, but stems from differences between dense and sparse matrix multiplication implementations on GPU. Since the two have the same complexity in theory ($O(mkn)$ for matrices of $m\times k$ and $k\times n$), this difference could be due to nuanced hardware (eg GPU) and software (eg CUDA) reasons (explained more in A3 below). Overall, we believe that it is still fair to say that dense and sparse retrieval has similar compute as long as $k$ is relatively small (e.g. $k\leq 64$), which is exactly the region that one wants to use efficient embedding methods for fast retrieval -- and CSR could outperform MRL by very large accuracy margins in these regions, even beating MRL with much larger $k$. We will elaborate this discussion in the revision.
> > >
> > > ---
> > > **Q2.** About Figure 3a: Is the discrepancy among the timings (between MRL and CSR, and then also the timing improvement of CSR as changes) purely a result of efficient sparse matrix operations in PyTorch, or is it a difference in MRL vs CSR, or a difference in the way the timings were recorded for each method?
> > >
> > > **A2**. Indeed, this discrepancy is merely a result of PyTorch's sparse/dense matrix operations, not from the MRL/CSR methods themselves. We evaluate it following exactly the same recording protocol for fair comparison. The main goal of this figure is exactly to get rid of these hardware nuances by benchmarking the runtime of sparse and dense operators under the same active dimension. The figure shows that the two have roughly the same time, which facilitates us to focus on the complexity measure of ``active dimension`` when comparing MRL and CSR. We will further clarify this in the discussion of Figure 3a as well. Thank you for noting this.
> > >
> > > ---
> > > **Q3** I would happy to raise the score if there is a more definitive explanation why being more sparse improves the timings.
> > >
> > > **A3.** Thanks! As discussed above, this difference in timing is caused by the implementation of dense/sparse matrix operations in PyTorch and GPU. While a rigorous analysis of this issue is actually beyond the scope of our work and ICML, we do look deeply into this problem in these days, and as a result, we have a preliminary insight of this problem that might be helpful for understanding this difference.
> > >
> > > Remind that for sparse multiplication, only the **overlapped** non-zero elements contribute to the outcome. For example, for calculating the $ij$-th output, we only need to use the overlapped activations in sparse vectors $s_i,s_j$. If no overlap at all, we can even omit it. Examining overlap only requires **comparing indices (int)** in the sparse matrices, which is much faster than **multiplying float vectors**. In a very sparse matrix (eg if the dimension $h$ is very large), the overlap could be more rare, leading to an even faster retrieval. This is a very nice property of CSR: it means that ***we can use a larger embedding dimension $h$ (more information) while achieving an even faster retrieval in the same time***! In comparison, in MRL/dense embedding, a larger dimension always leads to slower retrieval.
> > >
> > > To verify this in practice, we benchmark the number of multiplications using both dense and sparse matrices in CSR format (with row-wise product [1]) under the same default setup.
> > >
> > > |Active Dim|MRL| CSR (h=8192) | CSR (h=16384) | CSR (h=32768)|
> > > |-|-|-|-|-|
> > > |2|1.3×10e9|3.2×10e5|1.7×10e5|8.4×10e4|
> > > |4|2.6×10e9|1.3×10e6|6.7×10e5|3.4×10e5|
> > >
> > > We can see that the operation number of sparse ones can be several orders of magnitude smaller than that of the dense methods. Besides, a larger $h$ does further bring fewer computation and lead to faster retrieval in practice, which verifies our analysis above.
> > >
> > >
> > > Besides, there could also be other nuanced factors. For example, in PyTorch, sparse and dense  multiplications call different backends: dense ones use cuBLAS GEMM that is highly optimized but heavyweight, while sparse ones uses cuSPARSE that has lower launch overhead. As this difference is more system-related, we leave more comprehensive analysis for future work.
> > >
> > > We will add this discussion for a well-rounded understanding.
> > >
> > >
> > > [1] Gustavson. "Two fast algorithms for sparse matrices: Multiplication and permuted transposition." ACM Transactions on Mathematical Software, 1978
> > >
> > > ---
> > >
> > > Hope the elaboration help alleviate your concerns and please let us know if there is more to clarify!

---

### Official Review · Reviewer_PD5c · 2025-03-15

**Overall Recommendation:** 3

**Summary:**

In this paper, the authors propose Contrastive Sparse Representation (CSR) as an alternative to Matryoshka Representation Learning (MRL) for adaptive embeddings. MRL requires retraining models and suffers from performance drops at shorter embedding lengths, while CSR achieves adaptive representation through sparse coding, preserving high-dimensional semantic quality. CSR combines reconstruction-based sparse autoencoding with a contrastive loss to maintain accuracy and retrieval efficiency at various sparsity levels. Experiments on image, text, and multimodal benchmarks show that CSR outperforms MRL in accuracy and retrieval speed while requiring significantly less training time (up to 69× faster). CSR maintains the semantic integrity of the original embeddings and achieves strong generalization across downstream tasks with fewer computational resources, and it is a more efficient and scalable approach for adaptive representation learning.

**Claims And Evidence:**

CSR is shown to achieve higher performance and speed than MRL, supported by extensive experiments on ImageNet, MTEB, and MS COCO. The reduction in training costs is also well-supported, with experiment results showing that CSR requires significantly less training time than MRL, achieving up to a 69× speedup on ImageNet1k tasks. Furthermore, the paper provides evidence that CSR preserves semantic quality while improving efficiency by using a reconstruction-based sparse coding approach with contrastive loss, as shown through consistent accuracy at different sparsity levels.
The generalization claim across modalities is overstated since the multimodal experiments are limited to MS COCO and Flickr30K, which may not represent broader multimodal challenges

**Essential References Not Discussed:**

No, the paper discusses the key related works thoroughly.

**Experimental Designs Or Analyses:**

Yes - the experiment designs and analyses are sound and well-constructed. The authors evaluate CSR across multiple benchmarks (ImageNet, MTEB, and MS COCO) and provide thorough comparisons with MRL and other baselines using consistent and relevant metrics like retrieval accuracy, inference time, and training costs. The analysis includes ablation studies, scaling experiments, and tests on different sparsity levels. The inclusion of both vision and text tasks, along with multimodal settings, adds further credibility to the evaluation.

**Methods And Evaluation Criteria:**

The proposed methods and evaluation criteria are clearly defined and appropriate for the problem. The authors present a detailed explanation of the CSR framework and evaluate it on a range of benchmarks (ImageNet, MTEB, MS COCO) using relevant metrics like retrieval accuracy and inference time. The comparison with MRL and other baselines is fair and well-structured.

However, I believe the authors can do a better job at highlighting the novelty of the work. A lot of material in Section 3.2, especially Section 3.2.2, would fit better in the Preliminaries or Related Work section. This would help keep the focus on the core contributions of the paper, preventing the audience from being distracted by too much technical detail before understanding the main idea.

**Other Comments Or Suggestions:**

Providing a brief discussion on potential limitations or failure cases of CSR would further strengthen the paper.

**Other Strengths And Weaknesses:**

The paper does not explore how CSR handles complex multimodal tasks beyond simple retrieval tasks (e.g., cross-modal generation or reasoning).
The scalability of CSR on extremely large datasets (beyond ImageNet or MS COCO scale) is not tested, which limits its generalizability to real-world large-scale applications.
The impact of different backbone architectures (e.g., transformer vs. convolutional models) on CSR’s performance is not fully examined.

**Questions For Authors:**

Can you clarify how CSR performs under extreme sparsity constraints (e.g., TopK = 4 or 8)? While the paper shows strong performance at moderate sparsity levels, it would be helpful to see a more detailed analysis of how CSR handles extreme sparsity, especially in comparison to MRL.

**Relation To Broader Scientific Literature:**

The paper builds on prior work in adaptive representation learning, particularly MRL, but proposes a more efficient sparse coding approach using reconstruction and contrastive learning. It also draws from earlier approaches of sparse autoencoders and contrastive learning to improve retrieval accuracy and efficiency while reducing training costs.

**Theoretical Claims:**

N/A - there is no theoretical claims.

---

> ### Author Rebuttal · Authors · 2025-04-01
>
> Thanks for your careful reading and critical review. Following your suggestions, we have added more discussions on complex multimodal generation ability and the scalability of CSR. We further address each of your concerns below and hope you find them satisfactory.
>
> ---
>
> **Q1** Move extensive technical details (especially Section 3.2.2) to the Preliminaries or Related Work section to highlight the paper's core innovations.
>
> **A1.** We sincerely appreciate this suggestion. As you suggested, we will move the background part of Sec 3.2.2 to Preliminaries for better readability. In this way, Sec 3.2 would be mostly devoted to the design of CSR and highlight the novelty of our design.
>
> ---
> **Q2** CSR's ability to handle complex multimodal tasks(e.g., cross-modal generation).
>
> **A2.** In this work, our primary goal is to develop a more efficient approach to representation learning. To evaluate the quality of learned representations, we follow the standard evaluation protocol in the efficient representation learning literature, such as MRL [1], focused mainly on well-known image (e.g., ImageNet) and text embedding (MTEB [2]) benchmarks.
>
> We further evaluate CSR on multimodel embeddings as well in Table 2 (in the main text) and *Table 1* below (zero-shot retrieval performance) and follow common evaluation protocols of CLIP embeddings (e.g.,  CLIP\_benchmark [3]) with standard metrics like 1-NN accuracy, NDCG@10, and Recall@5.
> Although CLIP embeddings can be used for multiple downstream tasks (including generation), our focus here is not to explore these alternatives, and thus we follow the standard protocol of CLIP evaluation for our experiments. Please let us know if you need any further clarification.
>
> Ref:
>
> [1] Kusupati, Aditya, et al. "Matryoshka representation learning." NIPS,2022.
>
> [2] Muennighoff, Niklas, et al. "MTEB: Massive text embedding benchmark." arXiv preprint arXiv:2210.07316 (2022).
>
> [3] https://github.com/LAION-AI/CLIP_benchmark
>
> ---
> **Q3** The scalability of CSR on extremely large datasets (beyond ImageNet or MS COCO scale) is not tested, which limits its generalizability to real-world large-scale applications.
>
> **A3.** Our evaluation on ImageNet-1k follows the standard setup in MRL, which also did not have results that far beyond this scale as well. Indeed, **with constraints of our academic compute**, ImageNet-1k is already a large-scale dataset for us and it is not really feasible for us to conduct industry-level "extremely large datasets" like JFT-300M. While MRL only evaluates image and text scenarios, we are the first to include the evaluation on multimodal embeddings as well, showing consistent gains as the other domains. Therefore, we believe that our results (marked as ``comprehensive`` by Reviewers 11DH and cpsE) do support the generality of our approach.
>
> Following your suggestion, we evaluated CSR against MRL on the larger CC3M dataset (3M images, compared to ImageNet's 1M and MS COCO's 0.3M). Results in Table 1 demonstrate CSR's consistent superiority across various active dimensions, confirming its scalability.
>
> *Table 1: Zero-Shot Retrieval Performance on MS COCO*
> |Model|Active Dim|I2T@5|T2I@5|
> |-|-|-|-|
> |ViT-B/16(Pre-trained)|512|69.23|83.03|
> |+MRL|256|54.46|61.06|
> |+CSR|256|**57.75**|**70.34**|
> |+MRL|128|48.96|55.86|
> |+CSR|128|**49.97**|**63.12**|
> |+MRL|64|38.71| 45.72|
> |+CSR|64|**40.19**|**52.39**|
>
> ---
> **Q4** The impact of different backbone architectures (e.g., transformer vs. convolutional models) on CSR’s performance is not fully examined.
>
> **A4.** In fact, we have included the results of CSR under both transformer (ViT) and convolutional models (ResNet-50) in both Figure 4 and Figure 5, where CSR behaves quite similarly under different backbone architectures, indicating that CSR is rather general and agnostic of the underlying backbone architecture.
>
> ---
> **Q5** Providing a brief discussion on potential limitations or failure cases of CSR.
>
> **A5.** CSR faces dead latent issues under extreme sparsity, particularly in multimodal settings(Sec. D.4). Based on our extensive experiments in other domains and ablation studies, we identify this as a technical challenge requiring solutions such as alternative loss designs or deeper architectures.
>
> ---
> **Q6** Can you clarify how CSR performs under extreme sparsity constraints (e.g., TopK = 4 or 8)? While the paper shows strong performance at moderate sparsity levels, it would be helpful to see a more detailed analysis of how CSR handles extreme sparsity.
>
> **A6.** Good question! Our additional experiments (Table 2 below) demonstrate CSR's robust performance even under extreme sparsity constraints (TopK = 2 or 4), consistently outperforming MRL
>
> *Table 2: 1-NN results on ImageNet 1k*
> |Active Dim|2|4|8|16|
> |-|-|-|-|-|
> |CSR|66.17|69.97|73.84|74.39|
> |MRL|-|-|62.19|67.91|
>
> Thank you for your constructive feedback. We hope our responses have addressed your concerns, and we welcome any further questions or discussion.

---

### Official Review · Reviewer_cpsE · 2025-03-17

**Overall Recommendation:** 3

**Summary:**

The paper presents Contrastive Sparse Representation (CSR) as a novel approach to adaptive representation learning, addressing the limitations of Matryoshka Representation Learning (MRL), which requires extensive retraining and suffers from performance degradation at shorter representation lengths.

**Claims And Evidence:**

Claim 1: CSR Outperforms MRL in Accuracy and Speed
Evidence: Compared to the MRL method, CSR achieves average performance gains of 4.6% and 6.8% on image-to-text retrieval, and 9.1% and 6.5% on text-to-image retrieval across the two datasets. This indicates that CSR provides higher accuracy while maintaining efficient retrieval times.
Claim 2: CSR Reduces Training Time Significantly
Evidence: The training time for CSR is reported to be a fraction of that required by MRL. For instance, CSR can be trained on ImageNet in about half an hour with a single GPU, compared to the extensive retraining needed for MRL.

**Essential References Not Discussed:**

no

**Experimental Designs Or Analyses:**

The experimental designs and analyses in the paper are largely valid and appropriate for evaluating CSR.
However, improvements could be made in terms of documentation for reproducibility, broader comparisons with other methods, and the inclusion of statistical significance testing.

**Methods And Evaluation Criteria:**

The proposed methods and evaluation criteria are well-aligned with the problem of adaptive representation learning. CSR effectively addresses the limitations of previous approaches, and the chosen metrics provide a comprehensive assessment of its performance across multiple dimensions. This makes the methodology robust and applicable to real-world scenarios, thereby contributing valuable insights to the field.

**Other Comments Or Suggestions:**

no

**Other Strengths And Weaknesses:**

no

**Questions For Authors:**

no

**Relation To Broader Scientific Literature:**

no

**Theoretical Claims:**

Several theoretical claims are made regarding the effectiveness and efficiency of Contrastive Sparse Representation Learning (CSR). However, the paper does not provide formal proofs for these claims.

---

> ### Author Rebuttal · Authors · 2025-04-01
>
> We sincerely thank you for your thoughtful assessment of our paper. We appreciate the recognition of our work's contributions, particularly noting the clear improvements of CSR over MRL by ``Outperforms MRL in Accuracy and Speed Evidence``, ``Reduces Training Time Significantly``.
>
> Meanwhile, thank you for acknowledging that our ``methods and evaluation criteria are well-aligned with the problem of adaptive representation learning`` and provide ``comprehensive assessment of its performance across multiple dimensions``.
>
> Next,  we will address your comments on the aspects for further improving this work.
>
> ---
>
> **Q1** Documentation for reproducibility.
>
> **A1.** Experimental details with key hyperparameter settings are provided in Appendix B.3, C.4, and D.3. We have included preliminary ([code](https://anonymous.4open.science/r/ICML_rebuttal-78D1/)). and will definitely complete the resources available for reproducibility.
>
> ---
> **Q2**. Broader comparisons with other methods.
>
> **A2.** Thanks for your suggestions! We acknowledge that several other methods (e.g., pruning, quantization, and distillation) also enable acceleration.
> However, it is important to clarify that while these methods primarily focus on accelerating the backbone model and embedding generation, CSR distinctively focuses on optimizing the post-processing phase, specifically the transition from embedding to retrieval. This distinction positions CSR as fundamentally **orthogonal** to existing acceleration approaches.
>
> To illustrate, we combine CSR with Int8 quantization, as demonstrated in the *Table 1* below. This combination achieves additional acceleration beyond quantization alone while incurring only minimal degradation in the compressed model’s performance.
> The effectiveness of CSR arises from its capability to maintain high performance even when employing significantly reduced active dimensions (e.g., 8). Compared to alternative approaches, applying CSR to the original model consistently results in the highest performance retention.
> We will incorporate this detailed discussion into the revised manuscript.
>
> *Table 1: 1-NN Acc Comparison on Different Methods*
>
> | Method    | Active Dim | Vanilla | Int8 Quant | Retrieval Time |
> |-----------|------------|---------|------------|----------------|
> | Resnet50  | 2048       | 75.19   | 73.48      | 5.17           |
> | +CSR      | 8          | 73.84   | 72.32      | 0.28           |
> | Resnet50  | 2048       | 70.97   | 69.11      | 5.16           |
> | +MRL      | 8          | 62.19   | 59.38      | 0.42           |
>
> ---
> **Q3** Statistical significance testing.
>
> **A3.** Thank you for your suggestion! Due to time constraints, we firstly report the standard division results in *Table 2* below for the comparison under 8 active dimensions, calculated with 5 independent runs. We can see that the stdev is only around 0.5, while the gap between MRL and CSR can be as large as 15.9 (for MRL, we adopted the original paper's results that did not report stdev). Consequently, our method consistently outperforms MRL by a significant margin.
>
> *Table 2: CSR statistical results on ImageNet1k*
>
> | Methods | Active Dim | 1-NN Acc     |
> | ------- | ---------- | ------------ |
> | MRL     | 8          | 62.19        |
> | MRL-E   | 8          | 57.45        |
> | CSR     | 8          | 73.39 ± 0.51 |

---

### Official Review · Reviewer_cLDh · 2025-03-17

**Overall Recommendation:** 5

**Summary:**

The authors propose a method for converting pretrained dense embedding vectors into sparse embedding vectors and show that it often outperforms standard approaches such as Matryoshka Representation Learning (MRL) in terms of both accuracy, training time and retrieval speed.

Their CSR method is inspired by Sparse Autoencoders and projects fixed embeddings into a higher-dimensional space and activating only the TopK dimensions for a compact representation. CSR uses an SAE style reconstruction loss as well as a non-negative contrastive loss.

They compare across various domains, architectures and tasks. This is an exciting contribution to the field and has the potential to become the new standard for efficient vector representations.

**Claims And Evidence:**

The authors claim that Contrastive Sparse Representation (CSR)
1. Has higher accuracy than MRL when controlling for the number of active dimensions
* Figure 7(a)-(b) on ImageNet 1-NN accuracy
* Table 2 on MS COCO and Flickr30k (image to text and text to image)
* Figure 7(c) on a subset of the MTEB retrieval dataset
* Table 1 on a subset of data from MTEB
2. Has shorter training time than MRL
* Figure 1(c)
3. Has faster retrieval time than MRL
* Figure 1(b)
* Table 1 on subset of data from MTEB

**Essential References Not Discussed:**

No

**Experimental Designs Or Analyses:**

The authors have done a good job of benchmarking their approach across multiple modalities and benchmarks. They also benchmark against prior work (MRL by Kusupati et al. 2022)

**Methods And Evaluation Criteria:**

The proposed methods and benchmark datasets make sense for the problem at hand. They do a very thorough comparison against Matryoshka Representation Learning (MRL) by evaluating across similar domains and benchmarks used in the original MRL paper.

**Other Comments Or Suggestions:**

Typos:
Figure 1 caption “Compared to MLR” should be “MRL”
Figure 1 caption “we outperform MSR on 1-NN accuracy” should be “we outperform MRL”?
Section C.3 “MTEB benchmar” should be “benchmark”
Table 5 “NV-EmbeV2” should be “NV-EmbedV2”

The embedding models in Table 1 all use MRL; while this is stated clearly in the supplementary material, it would be helpful to state this explicitly in the main text.

My review is contingent on the authors releasing the code for CSR so that others can replicate their results.

**Other Strengths And Weaknesses:**

This work is potentially quite significant for the field of retrieval, text embedding and image embeddings.

**Questions For Authors:**

Figure 1 (c) how many active dims for the data points in this plot? Is this the same data as in Table 1? If so please state this explicitly.

In Figure 7(c), are the results for active dimensions 32? If so, please state this explicitly in the caption. Currently the caption hints at this by stating “the results of CSR-32…”.

In Figure 7(c), it would be helpful to see the full performance of each model (e.g. NV Embed, Nomic-v1.5 etc.) with the full vector representation. This would remind the reader that sparse representations with low active dimensions usually have lower accuracy than the full embedding.

It would be helpful to briefly discuss MRL-E, SVD and Rand-LP in the main text (and not just in the supplementary materials), as these appear in multiple figures and tables.

It would be helpful to elaborate on the Section E.3 with regards to retrieval time evaluation.

It is slightly confusing that the method is called CSR (contrastive sparse representation), while the embeddings are stored in the CSR (compressed sparse row) format. I assume this was intentional - if so, it could be worth explicitly mentioning this by saying something like “the training method is called CSR, and the vector is stored in the standard CSR format…”

**Relation To Broader Scientific Literature:**

The results in this paper are related to the broader literature around efficient vector representations. The main target of their approach, MRL, is widely used by modern embedding models including OpenAI’s text-embedding-3-large. Their approach has the potential to become the new standard for efficient representations.

**Theoretical Claims:**

The authors reference one theoretical proof for motivation from Wang et al. 2024, but do not rely on this for their experimental results. I did not check the correctness of this theorem.

---

> ### Author Rebuttal · Authors · 2025-04-01
>
> Thank you for your thoughtful review and for appreciating the quality of our work. The concerns have been addressed as below:
>
> ---
>
> **Q1** Typos in Figure 1 caption, Section C.3, and Table 5.
>
> **A1.** Thank you for pointing out! Following your great suggestions, we will fix them in the revised manuscript.
>
> ---
>
> **Q2** It would be helpful to state that embedding models in Table 1 use MRL.
>
> **A2.** Thank you for your great suggestion! We will clarify this distinction in Table 1 in the revised manuscript.
>
> ---
> **Q3** Releasing Codebase.
>
> **A3.** We will certainly open-source the entire project upon acceptance. For now, we have released the [codebase](https://anonymous.4open.science/r/ICML_rebuttal-78D1/) of the ImageNet results as a reference implementation.
>
> ---
> **Q4** How many active dims in Figure 1(c)?
>
> **A4.** Here, we computed the average 1-NN accuracy across active dimensions 8 to 128 for each method to have a holistic evaluation. We will add this explanation to the experiment setup in Appendix B for further clarification.
>
> ---
> **Q5** In Figure 7(c), are the results for active dimensions 32?
>
> **A5.** Thank you for pointing out! Indeed, the results in Figure 7(c) are for active dimensions 32. We will explicitly clarify this detail in the figure caption in the revised manuscript.
>
> ---
>
> **Q6** In Figure 7(c), it would be helpful to see the full performance of each model (e.g., NV Embed, Nomic-v1.5 etc.) with the full vector representation.
>
> **A6.** Thank you for pointing this out! We will include the full performance of each model for better reference. One major advantage of CSR compared to these MRL models is that it is a very lightweight method and can be easily built upon the latest SOTA embedding models, while the others have to rely on their own heavy pretraining, which leads to inferior full representation performance as well.
>
> Therefore, the advantages of CSR are essentially two folds: 1) we can use off-the-shelf SOTA embedding models for the best full representation performance, and 2) we have the slightest degradation when converted to efficient embeddings in a lightweight way.
>
> ---
> **Q7** Briefly discuss MRL-E, SVD and Rand-LP in the main text.
>
> **A7.** Thank you for this valuable suggestion! We will elaborate on their setups in the main text as well in the revision.
>
> ---
> **Q8** It would be helpful to elaborate on Section E.3 with regard to retrieval time evaluation.
>
> **A8.** Thanks for your valuable suggestion! Here is a more detailed breakdown of the evaluation protocol for the retrieval time:
> 1. We first precompute embeddings for the entire ImageNet training set, storing them in a standard CSR (compressed sparse row) format in GPU memory as the retrieval database.
> 2. We compute the retrieval time as the average over 2,000 retrieval rounds, following an initial warm-up period of 100 rounds. In each retrieval round, the query set consists of 512 samples randomly drawn from the database. The warm-up phase is standard practice when benchmarking GPU computations, as it effectively eliminates initialization bias.
>
> ---
> **Q9** It is slightly confusing that the method is called CSR (contrastive sparse representation), while the embeddings are stored in the CSR (compressed sparse row) format. I assume this was intentional - if so, it could be worth explicitly mentioning this by saying something like “the training method is called CSR, and the vector is stored in the standard CSR format…”
>
> **A9.** Indeed, we intentionally call our method CSR (contrastive sparse representation) to be the "ML version" of CSR, i.e., how to learn a model that converts dense embeddings to highly sparse ones. And thanks for your suggestions -- we will add the explicit discussions on this terminology connection and distinction to be clearer.
>
> ---
>
> We are genuinely grateful for your thoughtful feedback, which has been instrumental in helping us refine the manuscript. Please do not hesitate to reach out if you have any additional comments or questions.

---

### Official Review · Reviewer_11DH · 2025-03-17

**Overall Recommendation:** 4

**Summary:**

This paper focuses on the problem of creating adaptive representations from foundation models, focusing on contrastive sparse coding (CSR) as a novel method applied after pre-training to produce efficient representations for a range of downstream tasks. CSR is compared with Matryoshka Representation Learning (MRL), which creates dense representations at multiple scales by truncating dense feature vectors. In contrast to MRL, CSR creates sparse representations where a target number of dimensions activates per input. The paper demonstrates several advantages of CSR, including higher fidelity, faster retrieval, and lower training cost compared to MLR and other simple baselines. The experiments focus on image, text, and text-image tasks, showing consistent gains over MRL in all cases. Ablations analyze the design choices, including sparsity level, input and hidden dimensionality, and data scaling. The method seems practically useful as a simple post-training method for which.one can tradeoff computational efficiency with accuracy reasonably in some relevant downstream tasks.

**Claims And Evidence:**

The paper does a good job of comparing CSR to popular adaptive representation baselines within the MRL family along with some simpler baselines, though it may be useful to also compare with or at least discuss other approaches that focus on pruning, quantization, and/or distillation. The method shows clear gains over MRL in most cases examined across the sparsity spectrum both in terms of fidelity and runtime, though there is some loss in fidelity — the degree to which this matters is probably out of scope of this contribution, but is important to acknowledge.

**Essential References Not Discussed:**

N/A

**Experimental Designs Or Analyses:**

No specific concerns here. It could be useful to examine whether CSR holds in other representation learning settings beyond 1-KNN probes. What about linear probes and/or few shot scenarios?

**Methods And Evaluation Criteria:**

Methods and evaluation criteria appear comprehensive, and not designed to favor the CSR method from what I can tell. That being said, ablation studies could go deeper, e.g., to try to understand the limitations e.g. in fidelity at lower sparsity levels of the sparse coding approach, and whether wider hyperparameter searches across design factors may help reduce approximation error further.

**Other Comments Or Suggestions:**

Since Theorem 5 is taken from Wang et al and is not a conribution of this paper, make sure to refer to it as such in the text anywhere it’s mentioned to minimize confusion.

**Other Strengths And Weaknesses:**

The application of modern sparse coding (with contrastive task regularization) to adaptive feature representations is not something I’ve seen, which could be of good practical relevance in efficiency research. That being said, there is not much in the way of theoretical or empirical insights from this paper.

**Questions For Authors:**

From Figure 3, it’s not clear to me what makes TopK=16 a “sweet spot”. In terms of retrieval time, it is the highest, though it is indeed the case that benefits over MRL baseline are maintained. Can you explain why this is considered a sweet spot?

**Relation To Broader Scientific Literature:**

The paper relates to recent sparse autoencoder findings on interpretability that gained some interest in the past year, and may serve to interest researchers beyond that particular use. There’s also a potential connection to other recent works on sparsity including sparse mixtures-of-experts. In addition, there is a wide literature on post-training methods including pruning, distillation, quantization, etc., that could be discussed given the practical importance to efficiency, especially on device!

**Theoretical Claims:**

The paper did not introduce any new theoretical claims, though they did make use of Wang et al, Theorem 5. Given that the empirical evidence backed up the theoretical claim, I do not have specific concerns here. Of course, additional theory characterizing the limits of this CSR method would be welcome!

---

> ### Author Rebuttal · Authors · 2025-04-01
>
> We appreciate your constructive comments and suggestions, which are helpful for us to improve the quality of our paper further. The concerns have been addressed as below:
>
> ---
> **Q1** May be useful to add a discussion on pruning, quantization, and distillation methods.
>
> **A1.** Indeed, CSR, pruning, quantization, and distillation all enable acceleration.
> However, while the other three methods accelerate the backbone and inference embedding generation, our approach (CSR) focuses on post-processing optimization from embedding to retrieval. Because of this, CSR is **orthogonal** to those other methods.
>
> For example, combining CSR with Int8 quantization, as shown in *Table 1* below, provides additional speed-up beyond quantization alone, with minimal loss in the compressed model’s performance. This benefit arises because CSR utilizes a significantly smaller number of active dimensions (e.g., 8) while maintaining overall effectiveness. Compared to other methods, applying CSR to the original approach results in the highest retention of performance. Thanks for bringing this up, and we will include this discussion in the revised version.
>
> *Table 1: 1-NN Acc Comparison on Different Methods*
>
> |Method| Active Dim | Vanilla | Int8 Quant | Retrieval Time |
> |-|-|-|-|-|
> |Resnet50| 2048|75.19|73.48|5.17|
> |+CSR|8|73.84|72.32|0.24|
> |Resnet50|2048|70.97|69.11|5.16|
> |+MRL|8|62.19|59.38|0.42|
>
> ---
> **Q2** Ablation study on fidelity at lower sparsity levels.
>
> **A2.** Please see A6.
>
> ---
> **Q3** Whether wider hyperparameter searches across design factors may help reduce approximation error?
>
> **A3.** Thank you for raising this point. CSR's primary influential parameters are the activation dimension $k$ and the hidden dimension $h$, both of which we analyze extensively in **Figure 5**.
> As for other standard training hyperparameters (learning rate, loss coefficients, etc.), Gao et al. [1] conducted extensive ablations in the context of SAEs, and we adopted their default configuration.
> We will include this clarification in the revised version.
>
> Ref:
>
> [1] Gao, Leo, et al. "Scaling and evaluating sparse autoencoders." arXiv preprint arXiv:2406.04093 (2024).
>
> ---
> **Q4** Further evaluation in linear probes and few-shot scenarios.
>
> **A4.** Good question! Following your suggestions, we have conducted additional experiments evaluating CSR with linear probing (*Table 2*) and few-shot learning (*Table 3*).
>
> *Table 2* shows the Top-1 accuracy of CSR under linear probing on ImageNet1K, while *Table 3* reports 1000-way {3-, 5-, 7-}-shot average performance across three test sets on ImageNetV2.
> We utilize the same backbone architecture as MRL [2] in the few-shot scenario for a fair comparison.
>
> *Table 2* compares the linear probing performance of CSR and MRL across varying active dimensions, demonstrating that CSR exhibits minimal performance degradation relative to MRL.
> A similar trend appears in *Table 3*, where CSR shows stronger robustness in few-shot classification.
> These additional results underscore CSR’s consistently high performance across different downstream tasks beyond 1-NN. We hope this addresses your concerns, and we welcome any further discussion.
>
> *Table 2: Linear probing performance comparison between different methods.*
>
> |Methods |Active Dim|Top-1 Acc|
> |-|-|-|
> |ResNet50|2048|80.59|
> |+CSR|128|79.76|
> |+CSR|32|78.94|
> |+CSR|8|78.60|
> | ResNet50|2048|76.80|
> |+MRL|128| 76.30|
> |+MRL|32| 75.03|
> |+MRL|8|66.63|
>
> *Table 3: Few-shot performance comparison between different methods.*
>
> | Methods  | Active Dim | 3-Shot | 5-Shot | 7-Shot |
> |-|-|-|-|-|
> |ResNet50|2048|0.57|0.62|0.65|
> |+MRL|8|0.52|0.55|0.57|
> |+CSR|8|0.56| 0.61|0.63|
>
> Ref:
>
> [2] Kusupati, Aditya, et al. "Matryoshka representation learning." NIPS,2022.
>
> ---
>
> **Q5** Referation to Wang et al's Theorem 5.
>
> **A5.** Thanks for the reminder. We will revise the statement and include proper reference to Wang et al.
>
> ---
>
> **Q6** What makes TopK=16 a “sweet spot”?
>
> **A6.** Thank you for highlighting this point. Here, we consider $k=16$ as the ''sweet spot'' because it attains the optimal balance between accuracy and efficiency. As demonstrated in Figures 4 and 5, $k=16$ outperforms smaller $k$ values (e.g., 8) while maintaining higher efficiency than MRL (Figure 3(b)). *Table 4* further confirms that $k=16$ is a strong trade-off point.
> Nevertheless, using a smaller $k$ remains an option for those who prioritize speed over performance.
>
> *Table 4: Performance vs Efficiency under Different Sparsity*
>
> |Top K||2|4|8|16|32|
> |-|-|-|-|-|-|-|
> |1-NN Acc|| 66.17|69.97|73.84|74.39|74.53|
> |Relative Retrieval Time||0.2|0.2|0.3|0.5|1.4|
>
> ---
>
> We are sincerely grateful for your valuable feedback, which has been instrumental in refining our manuscript. Should you have any additional comments or questions, please do not hesitate to contact us.

---

> > ### Comment · Reviewer_11DH · 2025-04-09
> >
> > Thanks for addressing my questions. I like the additional experiments on few-shot and linear probing, and some of the points for discussion. Therefore I raise my score and believe this to be an interesting approach for the community to know about.
> >
> > One point I want to clarify though:
> > I’m confused by Table 1 because there are two different rows for Resnet50 with 2048 dimensions with different accuracy and runtime numbers. I’m not sure why one is used to benchmark with vs without CSR and the other is used to benchmark MRL.
> > Can you clarify?

---

> > > ### Author Response · Authors · 2025-04-09
> > >
> > > Thank you very much for your thoughtful feedback and for improving your score. We’re glad to hear that you found the updated results satisfactory. Below, we address your remaining question:
> > >
> > > ---
> > >
> > > **Q1** I’m confused by Table 1 because there are two different rows for Resnet50 with 2048 dimensions with different accuracy and runtime numbers. I’m not sure why one is used to benchmark with vs without CSR and the other is used to benchmark MRL.
> > >
> > > **A1.** Thank you for pointing this out! As illustrated in Figure 2, CSR can be added **on top of any SOTA backbones**, whereas MRL requires training the backbone **from scratch** in order to learn adaptive representations.
> > >
> > > Therefore, in *Table 1* of the rebuttal, the two ResNet-50 entries correspond to these two paradigms. Specifically:
> > > - The first row uses a ResNet-50 backbone pre-trained via the timm library [1], which serves as the fixed SOTA baseline for evaluating CSR.
> > > - The second row uses the ResNet-50 backbone from the original MRL paper [2], which was trained from scratch. We directly adopted their released weights for this comparison.
> > >
> > > More implementation details can be found in Sec B.3.
> > >
> > > To further clarify, we also include an additional experiment in *Table 5* below, where both CSR and MRL are applied on **the same backbone weights**. Even in this matched setting, CSR outperforms MRL by **a notable margin (+5.59)**. Given that CSR also requires significantly less training time (see Figure 1(c)), we believe this highlights CSR’s effectiveness and efficiency in learning adaptive representations. Thanks for your valuable comments! We will include these additional results in the revision.
> > >
> > > *Table 5: 1-NN results of different methods with same backbone weights*
> > > |Method|Active Dim| Vanilla | Int8 Quant | Retrieval Time |
> > > |-|-|-|-|-|
> > > |Resnet50|2048|70.97|69.11|5.16|
> > > |+CSR|8|**67.78**|**65.44**|0.28|
> > > |+MRL|8|62.19|59.38|0.42|
> > >
> > > Ref:
> > >
> > > [1] https://huggingface.co/timm/resnet50d.ra4_e3600_r224_in1k
> > >
> > > [2]Kusupati, Aditya, et al. "Matryoshka representation learning." NIPS,2022.
> > >
> > > ---
> > > **Q2** Run number difference between two ResNet50.
> > >
> > > **A2.** Thank you for taking the time to carefully review our rebuttal!
> > >
> > > In *Table 1*, we report **relative retrieval time**, consistent with the main paper. For additional clarity, we also provide the **raw absolute timing values** here: 0.014476 vs. 0.014448. The difference only appears at the fifth decimal place, which we attribute to **GPU-level randomness** rather than any difference in evaluation methods.
> > >
> > > Overall, both methods operate at the same scale. Importantly, CSR with an active dimension of 8 can **significantly improve efficiency on top of this baseline**, while maintaining strong performance.
> > >
> > > We appreciate you pointing this out, and we will include the raw timing values for all methods in the revision to improve clarity.
> > >
> > > ---
> > > We hope the above explanations address your concerns. Please don’t hesitate to let us know if any further clarification would be helpful.

---

### Decision · Program_Chairs · 2025-05-01

**Decision:**

Accept (oral)

**Comment:**

In this work, the authors present a new optimization method for learning adaptive embedding representations in foundational models. Their new method termed Contrastive Sparse Coding (CSR) is compared to strong baseline methods such as Matryoshka Representation Learning (MRL) and demonstrates gains in terms of overall fidelity, and lower training cost compared to MLR and other simple baselines. The experiments focus on image, text, and text-image domains. The reviewers were generally impressed with the strong empirical results and connection to previous lines of work in machine learning. The reviewers did note some concerns about not testing on complex multimodal settings, and scalability on large-scale datasets, however these concerns were largely resolved during discussion. Given the strong empirical results and wide potential application, this paper will be accepted for this conference.